# TIME-SENSITIVE WEIGHT AVERAGING FOR PRACTICAL TEMPORAL DOMAIN GENERALIZATION

## ABSTRACT

Temporal Domain Generalization (TDG) is a valuable yet challenging task that requires models to support temporal distribution shifts without access to future samples. Prior work utilized time-sensitive models that take timestamps as input or directly estimated optimal model parameters for each temporal domain. However, these methods were evaluated in oversimplified settings that do not scale to complex scenarios. To fundamentally enhance TDG's value for real-world applications, we propose three key principles for TDG method design: 1) Time-sensitive model, 2) Generic method, and 3) Realistic evaluation. Reflecting these guidelines, we propose **T**ime-sensitive **W**eight **A**veraging (**TWA**), a simple yet effective approach to apply weight averaging (WA) of specialists for every temporal domain. For principle 1), we train a selector network to estimate the good coefficients to average weights based on timestamp input. For principle 2), TWA is inherently generic, as WA requires no modification to model architecture. For principle 3), we incorporate more realistic benchmarks into TDG, including CLEAR-10, CLEAR-100, Yearbook, and FMoW-Time, which feature complex data distributions and natural temporal shifts. Extensive experiments conducted on these benchmarks demonstrate the practical value of TWA, *e.g.*, on CLEAR-10/100, TWA consistently improves accuracy over the baselines by up to $4\%$. We also demonstrate TWA boosts performance on common TDG benchmarks used in prior work. Lastly, we provide theoretical insights behind the outstanding performance of TWA.

## 1 INTRODUCTION

Many researchers have looked into solutions for addressing distribution shifts that occur between training and testing data, including Domain Adaptation (DA) (Hoffman et al., 2017; Saenko et al., 2010b) and Domain Generalization (DG) (Gulrajani & Lopez-Paz, 2021; Li et al., 2018b; Saenko et al., 2010a). They have proposed approaches that either leverage unlabeled data in the target domain or use only source domains. Others have considered predicting the distribution shifts instead. For example, in Temporal Domain Generalization (TDG) and related tasks, researchers assume a smooth shift in the training distribution over time (*e.g.*, ying Bai et al. (2022); Ke et al. (2021b;a); Lin et al. (2022); Mancini et al. (2019); Nasery et al. (2021); Ortiz-Jiménez et al. (2019); Srinivasan et al. (2022); Verwimp et al. (2022); Wang et al. (2020); Zeng et al. (2023)). They assumed that understanding say how a mobile phone's appearance has changed in the past may help predict future changes. To date, however, exploration in this direction remains limited. As illustrated in Fig. 1(a), prior work has either overlooked the temporal shift, or relies heavily on time-sensitive mechanisms at the expense of generalization (ying Bai et al., 2022; Nasery et al., 2021; Wang et al., 2020; Mancini et al., 2019). While time-sensitive mechanisms help models generalize to the future by modeling the temporal shift with architecture modifications and special operations, they also make it challenging to use large models trained on lots of data (*e.g.*, OpenAI (2023); Touvron et al. (2023); Kirillov et al. (2023)) that have shown good performance on domain-aware tasks (*e.g.*, Iwasawa & Matsuo (2021); Kim et al. (2022)). In addition, as our experiments show, they do not generalize to more complex settings that are more representative of many applications of TDG.

In this paper, we identify three key principles in designing a good TDG method: **1) Time-sensitive model**, **2) Generic method**, and **3) Realistic evaluation**. Under these guiding principles, we propose **T**ime-sensitive **W**eight **A**veraging (**TWA**), a weight averaging method that uses a selector

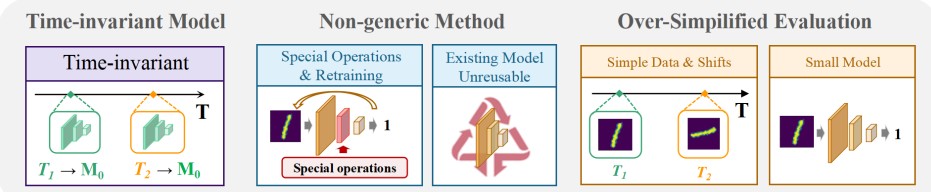

(a) The limitations of prior work (Cha et al., 2021; 2022; ying Bai et al., 2022; Nasery et al., 2021; Wang et al., 2020; Mancini et al., 2019).

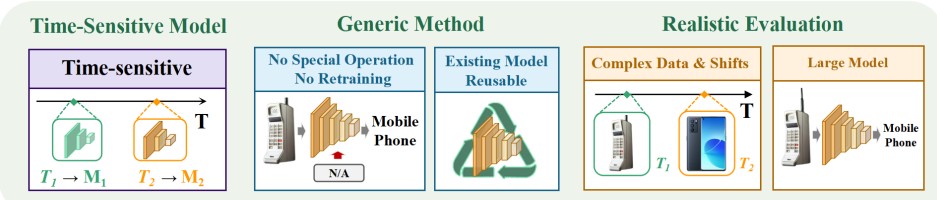

(b) Our TWA's three key design principles.

Figure 1: The 3 key principles of a good TDG method **1) Time-sensitive Model**: Model properties can vary temporally. **2) Generic Method**: The method can be easily combined with various architectures and tasks, requiring as few architecture modifications as possible. **3) Realistic Evaluation**: Evaluation settings need to be representative of real-world applications, e.g., complex data distributions, natural temporal shifts and large-scale models.

network optimized over source domains to estimate the optimal averaging coefficients for a corresponding temporal index, *i.e.*, a timestamp. As weight averaging makes no architecture assumptions, it is a generic method that can be easily incorporated into the training procedure of any model. Our selector network ensures that our approach is time-sensitive, *i.e.*, it accounts for the domain shifts at a specific temporal index. Finally, to ensure that our approach has practical value, we evaluate on more complex benchmarks than have been used in prior work on TDG. Specifically, we use CLEAR-10(Lin et al., 2022), CLEAR-100(Lin et al., 2022), Yearbook(Yao et al., 2022), and FMoW-Time(Yao et al., 2022), which feature complex data distributions and natural temporal shifts.

The closest work to ours are the Weight Averaging (WA) methods that have been explored in domain generalization (*e.g.*, Cha et al. (2021; 2022); Rame et al. (2022)). As our experiments show, WA outperforms prior work designed for TDG tasks (*e.g.*, Ortiz-Jiménez et al. (2019); Mancini et al. (2019); Wang et al. (2020); ying Bai et al. (2022); Nasery et al. (2021); Zeng et al. (2023)) despite lacking any sense of temporal shifts. To make WA temporally sensitive, we retain a number of specialized models and then learn to effectively compose them for a particular timestep, providing up to a 4% performance boost over WA.

Our contributions can be summarized as follows:
- We analyzed the current state of the art in Temporal Domain Generalization (TDG) from the perspective of real-world applications. We identify three key factors to this end: Time-sensitive model, Generic method, and Realistic evaluation.
- We propose Time-sensitive Weight Averaging (TWA), which imposes a time-sensitive selector network over weight averaging (WA) methods. TWA benefits from the generalizability of WA methods while being time-sensitive in its predictions. To the best knowledge of the authors, TWA is novel and has never been tried before. We also provide theoretical insights behind TDG tasks and our method.
- We propose an improved TDG benchmark by incorporating more realistic datasets: CLEAR-10, CLEAR-100, Yearbook, and FMoW-Time into the TDG setting. Extensive experiments demonstrate the superiority of our method, while also serving as baselines for future work.

## 2 RELATED WORK

**Domain Adaptation and Generalization.** Domain adaptation (DA) has been a long-standing research topic, and a large volume of work exists (Saenko et al., 2010b; Sun et al., 2015; Sun &

Saenko, 2016; Bousmalis et al., 2016; Hoffman et al., 2017; Gong et al., 2012; Tzeng et al., 2017; Liang et al., 2020; Ganin & Lempitsky, 2014; Long et al., 2017; Wang et al., 2018; Li et al., 2016; Bashkirova et al., 2023). However, DA methods require unlabeled data from the target domain. To address this, Domain Generalization (DG) methods trains models on one or multiple distinct source domain(s) that can generalize to any target domain without using data from target domains (*e.g.*,Li et al. (2017a); Muandet et al. (2013); Li et al. (2018a; 2017b); Gulrajani & Lopez-Paz (2021); Li et al. (2018b; 2019a)). Various DG methods have been proposed and can be broadly categorized into three groups: data manipulation, representation learning, and learning strategy design (Wang et al., 2022a). Weight Averaging (WA) (Cha et al., 2021; 2022; Rame et al., 2022; Wortsman et al., 2022) is an example of learning strategy design, which promotes the generalization capability with different strategies to average model weights. Another common learning strategy is meta-learning (Balaji et al., 2018; Li et al., 2017b; 2019b), which constructs meta-learning tasks to simulate domain shift.

**Temporal Domain Generalization** (TDG) (Ortiz-Jiménez et al., 2019; Mancini et al., 2019; Wang et al., 2020; ying Bai et al., 2022; Nasery et al., 2021; Zeng et al., 2023; Wang et al., 2022b) emerges as a specialized branch of Domain Generalization (DG) with two key distinctions: 1). Temporal domains in TDG can be continuously indexed over time; 2) TDG can leverage temporal information, e.g. timestamps, to help models generalize to future domains. In some earlier studies, TDG scenarios are encompassed within applicable scope of Continuous Domain Adaptation methods. CIDA (Wang et al., 2020) addresses continuously indexed domains by using the domain index as network input and adversarial DA to learn invariant representations. AdaGraph (Mancini et al., 2019) puts all domains on a graph and leverages the metadata of each domain to predict BatchNorm parameters. Recently, GI (Nasery et al., 2021) propose to enable a model to extrapolate into the near future by supervising the first-order Taylor expansion of the learned function. DRAIN (ying Bai et al., 2022) uses a Bayesian framework and recurrent graph generation to predict model parameters in the target domains. Most prior TDG works require architectural changes and use oversimplified evaluation settings, such as RotatedMNIST(Wang et al., 2020; ying Bai et al., 2022; Nasery et al., 2021).

**Continual Learning** primarily focuses on the "continual" aspect of the training paradigm. Most Continual Learning works (Zenke et al., 2017; Lopez-Paz & Ranzato, 2017; Shin et al., 2017; Chaudhry et al., 2018) place a greater emphasis on the "catastrophic forgetting" problem. Prior Continual Learning works also have oversimplified evaluation settings. Recent new datasets feature complex distributions and natural temporal shifts. CLEAR (Lin et al., 2022) is based on the YFCC100M dataset (Thomee et al., 2016) with a natural temporal evolution of real-world visual concepts that spans a decade. Wildtimes (Yao et al., 2022) also captures real-world temporal shifts, comprising 5 datasets for CV, NLP and Data Mining tasks. Yearbook and FMoW-Time are image datasets with categorical labels.

## 3 TIME-SENSITIVE WEIGHT AVERAGING FOR TEMPORAL DOMAIN GENERALIZATION

In Temporal Domain Generalization (TDG) a model must be able to account for shifts in the data distribution over time in order to improve task performance. A common assumption is that these distributions have smooth changes (*e.g.*, how the appearance of a mobile phone may change over time) that can be learned. More formally, we define a distribution function, $d(t)$ that shifts over time $t$. In theory, we can have infinite domains continuously distributed along the time axis: $\mathcal{D}(t) = \{(x_0, t, y_0), (x_1, t, y_1)...\}$, where $(x_i, y_i) \sim d(t)$. However, we cannot sample infinite temporal domains in practice, and many datasets discretely partition the data into finite domains $D_1, D_2, \ldots, D_E$ with timestamps $t_0 \leqslant t_1 \leqslant t_2 \leqslant \ldots \leqslant t_E$. Given a split timestamp $t_s$, we can divide all domains into source domains, $D_1, D_2, ..., D_s$, and target domains (future), $D_{s+1}, ..., D_E$. Given a network function $\mathcal{F}(\theta, \varphi)$, parameterized with model weights $\theta$ and generalization strategy $\varphi$, the goal of TDG is to find the optimal generalization strategy $\varphi^*$ to enable models trained on $D_1, D_2, ..., D_s$, $\mathcal{F}(\theta^*, \varphi)$, to generalize to $D_{s+1}, ..., D_E$, i.e. minimize the risk on target domains. Using $\mathcal{E}(\cdot, D_i)$ to denote the risk function on target domains $D_i$ and $\mathcal{E}_\varphi(\cdot, D_j)$ to denote the objective function within strategy $\varphi$ on domain $D_j$, we can frame TDG as a bi-level optimization problem:

$$\varphi^* = \arg\min_\varphi \sum_{i\in[s+1,E]} \mathcal{E}(\mathcal{F}(\theta^*, \varphi), D_i), \quad \text{s.t.} \quad \theta^* = \arg\min_\theta \sum_{j\in[1,s]} \mathcal{E}_\varphi(\mathcal{F}(\theta, \varphi), D_j). \quad (1)$$

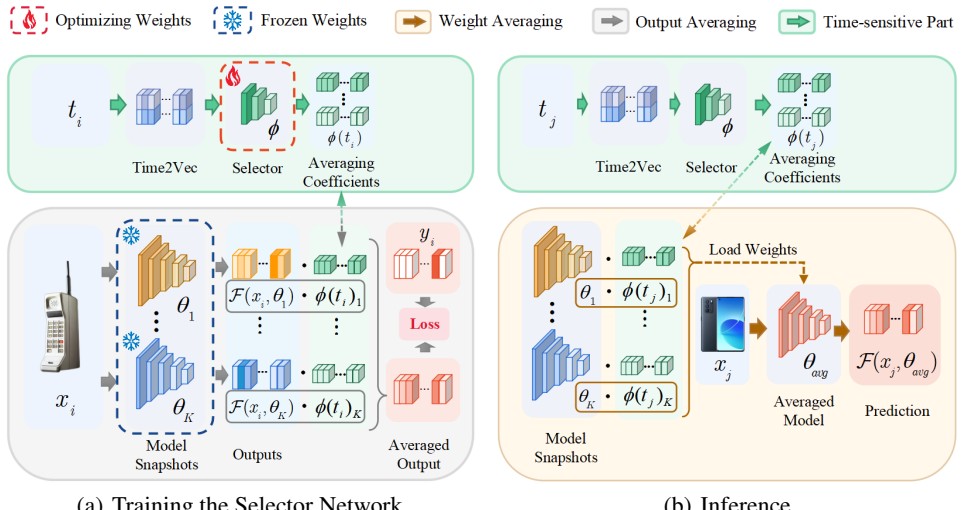

(a) Training the Selector Network         (b) Inference

Figure 2: An overview of our TWA method. (a) When optimizing the selector network in TWA, we use output averaging as a proxy task, utilizing the estimated coefficients to average the outputs of all snapshots. (b) During inference, we perform weight averaging with the optimized selector network.

To address this task, we propose Time-sensitive Weight Averaging (TWA), which uses a selector network to choose how to combine domain specialist models for new temporal domains. Sec. 3.1 discusses how we adapt weight averaging to better support temporal domain shifts. Sec. 3.2 describes our selector network, and Sec. 3.3 provides a theoretical analysis of our approach. Fig. 2 gives an overview of our model, and Sec. F in the appendix we provide detailed pseudo-code for training TWA.

### 3.1 ADOPTING WEIGHT AVERAGING FOR TDG

Weight Averaging (WA) methods have demonstrated good domain generalization performance without introducing new network components in prior work (*e.g.*, Cha et al. (2021; 2022); Rame et al. (2022)). Given sampling function $\mathcal{S}(\cdot)$ that samples $K$ model snapshots (training iterates) $\theta_{1:K} = [\theta_1, ..., \theta_K]$, a weight-average is $\theta_{avg} = \theta_{1:K} \cdot \mu_{1:K}$ with averaging coefficients $\mu_{1:K} = [\mu_1, ..., \mu_K] \in \Delta^K$. SWAD (Cha et al., 2021) is a WA method that densely averages weights from all iterations in an "overfit-aware" sampling zone determined via validation loss. This results in flatter minima and smaller domain generalization gaps.

Our task differs from SWAD's because we assume the domain-shift is temporally aligned and smooth. In our experiments, we found that optimizing BatchNorm (BN) layers in the models was necessary for good performance (6% drop on CLEAR-10 otherwise), which are often kept frozen on DG benchmarks (Gulrajani & Lopez-Paz, 2021). A natural fix is to post-update BN statistics in averaged models, but this becomes computationally expensive when using time-sensitive averaging coefficients, and we have observed that directly averaging BN statistics can result in model collapse when the sampling zone is too large.

To address these issues, we propose a simple trick, Late Sampling, in which we empirically set the size $I_{ls}$ of the sampling zone. Then, instead of sampling from $[I_s, I_e]$, we sample from $[\max(I_s, I_e - I_{ls}), I_e]$. In theory, using late sampling would require two training passes to reach the end of the sampling zone $I_e$ first, but we can only use one training pass in practice by estimating $I_e$.

### 3.2 PREDICTING TIME-SENSITIVE AVERAGING COEFFICIENTS USING A SELECTOR NETWORK

Weight averaging (WA) methods that have been proposed in prior work were applied to the domain generalization task (*e.g.*, Cha et al. (2021; 2022); Rame et al. (2022)). As our experiments will show, a direct application of these methods to the TGD task is effective, but contains one critical flaw: WA methods from prior work are not time sensitive. Thus, they are unable to take advantage

Table 1: Benchmark statistics and experiment setup for CLEAR(Lin et al., 2022), Yearbook(Yao et al., 2022) and FMoW-Times(Yao et al., 2022).

| Benchmark | # Sample | # Class | Split Timestamps | Input Shape | Models | Pretrain |
|---|---|---|---|---|---|---|
| CLEAR-10 | $35,000$ | 10 | $[t_9, t_8, t_7, t_5, t_3]$ | $(224, 224, 3)$ | R18, R50 | n/a, CB0 |
| CLEAR-100 | $150,000$ | 100 | $[t_9, t_8, t_7, t_5, t_3]$ | $(224, 224, 3)$ | R18, R50 | n/a, CB0 |
| Yearbook | $37,189$ | 2 | $[1980, 1970, 1960]$ | $(32, 32, 1)$ | R18 | n/a |
| FMoW-Times | $118,886$ | 62 | $[2017, 2016, 2015]$ | $(224, 224, 3)$ | R50 | ImageNet |

of the smooth changes to the distribution over time that is the hallmark of TGD. To address this, we propose to use the averaging coefficients $\mu_{1:K}$ to model the trajectory of the distribution shift over time. Specifically, given parameters of $K$ models $\theta_{1:K} = [\theta_1, ..., \theta_K]$, we define a "Selector Network" $\phi : \mathbb{R} \to \Delta^K$, so that $\phi(t)$ outputs a vector of averaging coefficients given timestamp $t$, $\mu_{1:K} = \phi(t)$. Thus, the WA formulation becomes:

$$\theta_{avg} = \theta_{1:K} * \phi(t) = \sum_{k=1}^{K} \phi(t)_k \cdot \theta_k \tag{2}$$

Our implementation of $\phi$ works by processing a normalized timestamp with a Time2Vec module (Kazemi et al., 2019) to capture high-frequency trends. Then we generate the coefficients via a small MLP. To choose models for averaging, we adopt a similar sampling strategy to the one proposed by SWAD (discussed at the end of Sec. 3.1). However, instead of using dense sampling strategy $\mathcal{S}_{ld}(\cdot)$, we use sparse reservoir sampling $\mathcal{S}_{ls}(\cdot)$ to choose $K$ snapshots, so as to limit the output size of $\phi$.

### 3.2.1 TWA TRAINING OBJECTIVE

For each temporal domain $D_i$, we train with objective $\hat{\mathcal{E}}(\cdot, D_i) = \sum_{(x,y,t)\sim D_i} \mathcal{L}(\mathcal{F}(x, \theta, t), y)$, where $\mathcal{L}$ is the selected loss function. We compute this empirical risk on source domains only, as we have no access to samples from target domains. And we optimize selector network $\phi$ to minimize the empirical risk on source domains.

Theoretically, we can derive the gradient of selector network parameters to optimize it directly. However, selector network is also used for averaging the BN statistics, which makes the implementation much more complex in practice, and our attempts to tackle this were unsuccessful. Hence, we propose to optimize the selector network using output averaging as a proxy task, which simplifies the implementation while still enabling efficient optimization. After the training process to optimize $\theta$, we add a few extra iterations to optimize selector network $\phi$. During inference, we use the optimized selector network to perform weight averaging. The final optimization problem becomes:

$$\phi^* = \arg\min_{\phi} \sum_{i \in [s+1,E]} \sum_{(x,t,y)\sim D_i} \mathcal{L}\left(\sum_{k=1}^{K} \phi(t)_k \cdot \mathcal{F}(x, \theta_k), \, y\right) \tag{3}$$
$$\text{s.t.} \quad \theta_{1:K} \sim \mathcal{S}_{ls}(\arg\min_{\theta} \sum_{j \in [1,s]} \sum_{(X,\cdot,Y)\sim D_j} \mathcal{L}(\mathcal{F}(X, \theta), Y))$$

### 3.3 THEORETICAL ANALYSIS OF TWA

In addition to our experiments, we have derived some theoretical guarantees on the ability of TWA to generalize from past to future data, given some mild assumptions about the nature of the temporal domain shift. We take inspiration from the theoretical work of SWAD (Cha et al., 2021), which found that models that live inside flat minima tend to generalize well. Our analysis shows that a *trajectory* of models that live inside flat minima w.r.t. their own temporal "domains" generalizes well into the future, provided that the domain shift is smooth and the space of selectors is well-constrained. Some empirical analysis confirms that TWA is indeed learning to produce weight-averaged models that stay within these flat minima as the data distribution shifts. Please see Supplementary D for a full exposition and proof of our theoretical results, as well as the empirical flat minima analysis.

Table 2: CLEAR-10 test accuracies ($\%$) on target domains using ERM, CIDA/PCIDA (Wang et al., 2020), AdaGraph (Mancini et al., 2019), SWAD (Cha et al., 2021) and our TWA.

| Model | Method | $D_{10}$ | $D_{9-10}$ | $D_{8-10}$ | $D_{6-10}$ | $D_{4-10}$ |
|-------|--------|----------|------------|------------|------------|------------|
| ResNet-18 | ERM (IID) | $86.7 \pm 0.6$ | $83.8 \pm 0.5$ | $84.3 \pm 1.5$ | $80.6 \pm 1.4$ | $76.3 \pm 0.9$ |
| | ERM (Last) | $86.3 \pm 2.0$ | $84.3 \pm 0.9$ | $82.5 \pm 1.5$ | $79.4 \pm 1.1$ | $76.6 \pm 0.8$ |
| | AdaGraph | $75,5 \pm 4.19$ | $74.8 \pm 1.9$ | $71.2 \pm 2.5$ | $56.4 \pm 3.5$ | $19.3 \pm 2.6$ |
| | CIDA | $85.6 \pm 0.5$ | $83.9 \pm 0.5$ | $81.3 \pm 0.9$ | $79.3 \pm 1.4$ | $71.5 \pm 4.3$ |
| | PCIDA | $87.3 \pm 0.8$ | $85.4 \pm 0.9$ | $82.9 \pm 1.0$ | $79.4 \pm 1.5$ | $72.4 \pm 2.2$ |
| | SWAD | $87.3 \pm 1.0$ | $86.0 \pm 0.6$ | $84.6 \pm 1.1$ | $81.5 \pm 1.4$ | $77.1 \pm 1.5$ |
| | **TWA (ours)** | $\mathbf{88.5 \pm 0.3}$ | $\mathbf{87.3 \pm 0.8}$ | $\mathbf{86.3 \pm 0.2}$ | $\mathbf{83.0 \pm 0.7}$ | $\mathbf{79.3 \pm 0.3}$ |
| ResNet-50 | ERM (IID) | $85.9 \pm 1.6$ | $84.9 \pm 1.0$ | $84.7 \pm 0.2$ | $81.0 \pm 1.9$ | $77.1 \pm 0.6$ |
| | ERM (Last) | $85.6 \pm 1.4$ | $85.0 \pm 1.8$ | $84.5 \pm 0.4$ | $80.6 \pm 1.0$ | $76.0 \pm 1.5$ |
| | SWAD | $87.3 \pm 0.9$ | $86.5 \pm 0.2$ | $85.8 \pm 1.1$ | $82.2 \pm 1.6$ | $78.5 \pm 1.1$ |
| | **TWA (ours)** | $\mathbf{89.0 \pm 0.3}$ | $\mathbf{87.8 \pm 0.3}$ | $\mathbf{86.9 \pm 0.2}$ | $\mathbf{84.0 \pm 0.5}$ | $\mathbf{80.3 \pm 0.4}$ |

Table 3: CLEAR-100 test accuracies ($\%$) on target domains using ERM, CIDA/PCIDA (Wang et al., 2020), AdaGraph (Mancini et al., 2019), SWAD (Cha et al., 2021) and our TWA.

| Model | Method | $D_{10}$ | $D_{9-10}$ | $D_{8-10}$ | $D_{6-10}$ | $D_{4-10}$ |
|-------|--------|----------|------------|------------|------------|------------|
| ResNet-18 | ERM (IID) | $68.3 \pm 0.3$ | $66.9 \pm 0.7$ | $64.9 \pm 1.4$ | $60.4 \pm 0.5$ | $53.3 \pm 0.9$ |
| | ERM (Last) | $67.0 \pm 1.1$ | $66.3 \pm 0.6$ | $64.4 \pm 0.5$ | $60.2 \pm 0.9$ | $53.1 \pm 0.8$ |
| | AdaGraph | $50.2 \pm 4.5$ | $39.5 \pm 2.5$ | $35.0 \pm 2.3$ | $21.0 \pm 2.6$ | $5.1 \pm 0.8$ |
| | CIDA | $67.8 \pm 0.3$ | $66.8 \pm 1.0$ | $66.5 \pm 0.3$ | $61.4 \pm 0.9$ | $52.7 \pm 1.2$ |
| | PCIDA | $69.2 \pm 0.1$ | $67.8 \pm 0.7$ | $67.2 \pm 0.7$ | $61.3 \pm 1.2$ | $53.3 \pm 1.7$ |
| | SWAD | $69.4 \pm 0.7$ | $67.3 \pm 0.7$ | $65.6 \pm 1.0$ | $61.5 \pm 0.8$ | $53.7 \pm 1.0$ |
| | **TWA (ours)** | $\mathbf{72.1 \pm 0.3}$ | $\mathbf{70.2 \pm 0.3}$ | $\mathbf{68.9 \pm 0.2}$ | $\mathbf{64.3 \pm 0.1}$ | $\mathbf{57.4 \pm 0.3}$ |
| ResNet-50 | ERM (IID) | $71.8 \pm 0.5$ | $69.1 \pm 0.4$ | $67.7 \pm 0.7$ | $63.5 \pm 0.3$ | $56.1 \pm 0.6$ |
| | ERM (Last) | $70.5 \pm 1.9$ | $68.5 \pm 1.3$ | $66.6 \pm 0.7$ | $63.3 \pm 1.3$ | $55.4 \pm 1.3$ |
| | SWAD | $72.1 \pm 0.5$ | $70.8 \pm 0.9$ | $69.0 \pm 0.9$ | $65.2 \pm 0.3$ | $57.8 \pm 1.1$ |
| | **TWA (ours)** | $\mathbf{75.1 \pm 0.6}$ | $\mathbf{73.4 \pm 0.4}$ | $\mathbf{72.0 \pm 0.3}$ | $\mathbf{68.3 \pm 0.1}$ | $\mathbf{60.4 \pm 0.7}$ |

## 4 EXPERIMENTS

Our experiments are designed to highlight two main capabilities of TWA: **Realistic Evaluation** and **Comprehensive Comparison**. On the former, newly incorporated benchmarks serve to showcase the practical value of our proposed TWA. The latter ensures that when making fair comparisons on the most commonly used TDG benchmarks to as much baseline methods as possible, TWA still demonstrates superior performance.

### 4.1 EXPERIMENTAL SETUP

**Evaluation Datasets.** We selected 7 datasets for evaluation. CLEAR10/100 (Lin et al., 2022), Yearbook (Yao et al., 2022), and FMoW-Times (Yao et al., 2022) are the newly incorporated datasets that are more challenging and reflect real-world applications. Rotated MNIST (Deng, 2012), Rotated Gaussian (Wang et al., 2020) and Portrait (Ginosar et al., 2015) are commonly used TDG datasets and we follow the settings in DDA (Zeng et al., 2023) for these 3 datasets.

- **CLEAR-10 and CLEAR-100** (Lin et al., 2022) contain user-uploaded images from 2007-2014 with natural temporal shifts of visual concepts. Samples are categorized into 10 bucket timestamps $t_1, .., t_{10}$. We select $t_9, t_8, t_7, t_5, t_3$ to split source and target domains, resulting in 5 settings with target domains $D_{10}, D_{9-10}, D_{8-10}, D_{6-10}, D_{4-10}$.
- **Yearbook** (Yao et al., 2022) contains American high school yearbook photos from 1930-2013, with changing social norms, fashion, and demographics, on which gender classification is conducted. Split timestamps are $1980, 1970, 1960$, so target domains are $1980 - 2013, 1970 - 2013, 1960 - 2013$.
- **FMoW-Times** (Yao et al., 2022) contains satellite images from 2002 to 2017 that capture temporal distribution shifts caused by human activity. The task is classifying land usage, and the split timestamps are $2017, 2016, 2015$, making the target domains $2017, 2016 - 2017, 2015 - 2017$.

Table 4: Yearbook test accuracies (%) on target domains, with ResNet-18 trained from scratch, using ERM, CIDA/PCIDA (Wang et al., 2020), SWAD (Cha et al., 2021), and our TWA

| Method | $1980 - 2013$ | $1970 - 2013$ | $1960 - 2013$ |
|---|---|---|---|
| ERM (IID) | $92.4 \pm 0.7$ | $83.3 \pm 0.8$ | $82.6 \pm 2.3$ |
| ERM (Last) | $92.0 \pm 0.8$ | $84.3 \pm 2.0$ | $81.7 \pm 0.8$ |
| CIDA | $92.2 \pm 1.2$ | $82.7 \pm 5.0$ | $75.0 \pm 5.4$ |
| PCIDA | $92.1 \pm 1.2$ | $85.6 \pm 3.9$ | $74.5 \pm 6.7$ |
| SWAD | $\mathbf{93.6} \pm \mathbf{0.7}$ | $85.3 \pm 1.7$ | $81.1 \pm 2.4$ |
| **TWA (Ours)** | $93.5 \pm 0.5$ | $\mathbf{85.7} \pm \mathbf{0.8}$ | $\mathbf{83.2} \pm \mathbf{0.9}$ |

- **Rotated MNIST** (Deng, 2012) is a semi-synthetic dataset where each MNIST image is rotated by a certain angle per domain.
- **Rotated Gaussian** (Wang et al., 2020) is a synthetic dataset with 30 domains generated by the same Gaussian distribution with rotating decision boundary.
- **Portrait** (Ginosar et al., 2015) is a simplified version of the Yearbook dataset, which contains photos of high school students over different years.

**Compared Baseline Methods.** We mainly compare our TWA with Domain Generalization (DG) and Temporal Domain Generalization (TDG) methods.

- **Domain Generalization (DG)** baseline methods include: **ERM**, **MTL**, **GROUPDRO** (Sagawa et al., 2019), **IRM** (Arjovsky et al., 2019), **MMD** (Li et al., 2018a), **CORAL** (Sun & Saenko, 2016), **SAGNET** (Nam et al., 2021), **SELFREG** (Kim et al., 2021), **SWAD (Cha et al., 2021) and MIRO** Cha et al. (2022). Specifically, SWAD and MIRO are time-agnostic WA methods from prior domain generalization works.
- **Temporal Domain Generalization (TDG)** baseline methods include: **MLDG** (Li et al., 2017b), **CIDA/PCIDA** (Wang et al., 2020), **AdaGraph** (Mancini et al., 2019), **EAML** (Liu et al., 2020), **LSSAE** (Qin et al., 2022), **DRAIN**(ying Bai et al., 2022), **GI** (Nasery et al., 2021),and **DDA** (Zeng et al., 2023). Note that CIDA, PCIDA, and AdaGraph are the most commonly used TDG baselines.We also provide a stand-alone comparison with DRAIN(ying Bai et al., 2022) in the appendix C.5 due to its highly task-specific design.

## 4.2 RESULTS

We first focus our experiments on the newly incorporated benchmarks. These experiments are designed to take into account three important aspects of TDG tasks: the extent of temporal distribution shifts, network architectures, and pre-training procedures. Prior work typically evaluated "near future" settings with only minor shifts (*e.g.* (ying Bai et al., 2022; Nasery et al., 2021)), where source and target domains are temporally "close", *e.g.*, $[D_1, ..., D_{29}] \rightarrow [D_{30}]$. In contrast, as shown in Tab. 1, we consider a larger variety of temporal shifts with different split timestamps.We employed two common backbones: ResNet-18 (R18) and ResNet-50 (R50) (He et al., 2016). These backbones were either pretrained on ImageNet (Deng et al., 2009), MoCo V2 (Chen et al., 2020), 700K unlabeled images from CLEAR (Lin et al., 2022) (denoted as CB0), or trained from scratch on source datasets. For all experiments, we report top-1 accuracy. See supplementary for additional details.

**Training from Scratch on CLEAR-10/100.** Tabs. 2& 3 report TDG performance on CLEAR-10 and CLEAR-100 datasets when training from scratch. Notably, the time-invariant Weight Averaging baseline (SWAD (Cha et al., 2021)) achieves up to a 5% boost in accuracy over PCIDA, the best prior TDG method in our comparison. That said, our time-sensitive WA approach, TWA, achieves an additional boost of 1-3%. We also find other time-sensitive methods, such as CIDA and PCIDA, perform well with slight temporal shifts. However, their performance degrades when faced with larger shifts, while our approach remains robust to these changes. Table 4 demonstrates that our approach also generalizes to the Yearbook dataset, where we see similar trends to those from CLEAR-100.

**Training with Pre-trained Weights on CLEAR-10/100.** The choice of pretraining method and data has significant impact on domain generalization performance (Iwasawa & Matsuo, 2021; Kim et al., 2022). Tab. 5 shows results on CLEAR datasets when pretrained on unlabeled data. Our approach still has a consistent advantage over SWAD, even when combined with MIRO (Cha et al., 2022). Notably, our biggest gains come on the more challenging CLEAR-100, where combining TWA with

Table 5: CLEAR test accuracies (%) on target domains, using R50 and CB0 pre-training, using ERM, SWAD (Cha et al., 2021), MIRO (Cha et al., 2022) and our TWA

| Dataset | Method | $D_{10}$ | $D_{9-10}$ | $D_{8-10}$ | $D_{6-10}$ | $D_{4-10}$ |
|---|---|---|---|---|---|---|
| | ERM (IID) | $92.3 \pm 1.1$ | $91.3 \pm 0.7$ | $91.6 \pm 1.0$ | $89.2 \pm 0.9$ | $85.9 \pm 0.7$ |
| | SWAD + ERM | $93.5 \pm 0.4$ | $93.0 \pm 0.3$ | $92.3 \pm 0.6$ | $90.1 \pm 1.0$ | $88.2 \pm 0.6$ |
| CLEAR-10 | **TWA + ERM** | $94.4 \pm 0.4$ | $93.8 \pm 0.3$ | $93.1 \pm 0.2$ | $91.2 \pm 0.3$ | $88.8 \pm 0.2$ |
| | MIRO (IID) | $95.3 \pm 0.5$ | $94.6 \pm 0.4$ | $93.9 \pm 0.9$ | $93.1 \pm 0.4$ | $90.9 \pm 1.1$ |
| | SWAD + MIRO | $96.3 \pm 0.2$ | $95.3 \pm 0.3$ | $94.9 \pm 0.7$ | $93.7 \pm 0.7$ | $91.5 \pm 1.0$ |
| | **TWA + MIRO** | $\mathbf{96.3 \pm 0.2}$ | $\mathbf{95.9 \pm 0.2}$ | $\mathbf{95.3 \pm 0.3}$ | $\mathbf{94.3 \pm 0.3}$ | $\mathbf{92.5 \pm 0.6}$ |
| | ERM (IID) | $68.3 \pm 0.3$ | $66.9 \pm 0.7$ | $64.9 \pm 1.4$ | $60.4 \pm 0.5$ | $53.3 \pm 0.9$ |
| | SWAD + ERM | $69.4 \pm 0.7$ | $67.3 \pm 0.7$ | $65.6 \pm 1.0$ | $61.5 \pm 0.8$ | $53.7 \pm 1.0$ |
| CLEAR-100 | **TWA + ERM** | $69.4 \pm 0.7$ | $67.3 \pm 0.7$ | $65.6 \pm 1.0$ | $61.5 \pm 0.8$ | $53.7 \pm 1.0$ |
| | MIRO (IID) | $81.7 \pm 0.1$ | $80.5 \pm 0.4$ | $79.1 \pm 0.5$ | $75.7 \pm 0.1$ | $70.7 \pm 0.1$ |
| | SWAD + MIRO | $83.0 \pm 0.7$ | $82.7 \pm 0.4$ | $80.1 \pm 1.4$ | $77.7 \pm 0.4$ | $73.0 \pm 1.1$ |
| | **TWA + MIRO** | $\mathbf{85.2 \pm 0.3}$ | $\mathbf{84.5 \pm 0.3}$ | $\mathbf{83.9 \pm 0.2}$ | $\mathbf{80.8 \pm 0.2}$ | $\mathbf{76.0 \pm 0.1}$ |

Table 6: FMoW-Times test accuracies (%) on target domains, with R50 and ImageNet pre-training, using ERM, SWAD (Cha et al., 2021), MIRO (Cha et al., 2022) and our TWA

| Method | $2017$ | $2016 - 2017$ | $2015 - 2017$ |
|---|---|---|---|
| ERM (IID) | $63.5 \pm 0.9$ | $50.1 \pm 1.6$ | $53.7 \pm 1.0$ |
| SWAD + ERM | $66.1 \pm 0.6$ | $48.9 \pm 0.6$ | $55.1 \pm 0.1$ |
| **TWA (ours) + ERM** | $\mathbf{69.5 \pm 0.9}$ | $\mathbf{51.5 \pm 0.3}$ | $53.8 \pm 0.3$ |
| MIRO (IID) | $64.5 \pm 1.9$ | $49.2 \pm 0.5$ | $54.9 \pm 1.4$ |
| SWAD + MIRO | $67.1 \pm 1.1$ | $49.9 \pm 1.3$ | $55.3 \pm 0.5$ |
| **TWA (ours) + MIRO** | $67.0 \pm 2.7$ | $51.0 \pm 1.5$ | $\mathbf{57.4 \pm 0.2}$ |

MIRO obtains 2-4% gains over prior work. Note that ImageNet pretraining is not suitable for TDG evaluation on CLEAR as there is significant overlap in the categories found in these datasets.

**Results on Yearbook and FMoW-Times.** Tab. 6 reports performance on the challenging FWoW benchmark. Variants of our TWA approach obtain best performance. Notably, although MIRO performed well on CLEAR, it did not generalize well to this setting. But our method is still able to boost performance over MIRO alone or combined with SWAD. The distributions and temporal shifts in Yearbook and FMoW-Times are more task-specific. Tab. 4 and Tab. 6 show that TWA is robust to diverse patterns of temporal shift.

**Comparisons in settings of prior work.** Many TDG methods do not generalize well to our more complex benchmarks. Thus, we follow the experimental settings of DDA (Zeng et al., 2023) to compare TWA and other DG and TDG methods over three commonly used TDG benchmarks. Tab. 7 reports our results, where combining TWA with DDA achieves state-of-the-art results on all three datasets. This demonstrates that TWA generalizes across many datasets and settings.

### 4.3 ABLATION STUDY

In this section, we run ablation experiments to answer two questions, "How Long?" and "How Many?", referring to the length of sampling zone and number of sampled snapshots. All the ablation experiments are conducted with ResNet-18 trained from scratch with a reduced training process for 10K iterations. Other experiment settings are the same as those in Sec. 4.2.

**How Long?** What is the adequate length of the sampling zone when applying WA method to TDG and optimizing BN layers? We take SWAD as an example, and analyze the impact of sampling zone length (in iterations) on test accuracy. We also evaluate the ERM performance when optimizing BN layers, ERM (IID), and the performance when freezing BN layers, ERM (FB). The results are shown in Fig. 3(a). We see that performance drops significantly when freezing BN layers or having a long sampling zone. It's safe to set the sampling zone length as 1000-3000 iterations.

**How Many?** What is the optimal number of sampled snapshots in our TWA? We analyze the impact of snapshot number on test accuracy of TWA. The results are shown in Fig. 3(b). It can be observed that TWA achieves improved performance as the number of snapshots increases. However, the rate

Table 7: Comprehensive comparisons with 2 synthetic benchmarks, Rotated MNIST (Deng, 2012) and Rotated Gaussian (Wang et al., 2020), and 1 real-world benchmark, Portrait (Ginosar et al., 2015). We compare our TWA with ERM, MTL, GROUPDRO, IRM, MMD (Li et al., 2018a), CORAL (Sun & Saenko, 2016), SAGNET (Nam et al., 2021), SELFREG (Kim et al., 2021), MLDG (Li et al., 2017b), CIDA (Wang et al., 2020), EAML (Liu et al., 2020), LSSAE (Qin et al., 2022), GI (Nasery et al., 2021), DDA (Zeng et al., 2023). The baseline results are from Zeng et al. (2023) if not noted.

| Dataset | Rotated MNIST | | | | Rotated Gaussian | Portrait |
|---|---|---|---|---|---|---|
| | 10° | 15° | 20° | 30° | | |
| GROUPDRO | 91.1 | 83.5 | 79.8 | 63.9 | $80.8 \pm 3.4$ | $92.6 \pm 0.2$ |
| IRM | 75.0 | 67.1 | 55.1 | 48.6 | $72.0 \pm 2.2$ | $91.3 \pm 0.4$ |
| MMD | 88.5 | 82.8 | 75.6 | 45.9 | $56.8 \pm 1.3$ | $92.0 \pm 0.2$ |
| CORAL | 91.9 | 84.1 | 77.6 | 63.2 | $56.8 \pm 1.1$ | $91.3 \pm 0.2$ |
| MTL | 92.8 | 84.1 | 77.5 | 63.1 | $56.4 \pm 1.4$ | $92.0 \pm 0.1$ |
| SAGNET | 91.9 | 86.8 | 79.2 | 62.6 | $52.0 \pm 1.8$ | $92.7 \pm 0.2$ |
| SELFREG | 93.0 | 87.5 | 77.9 | 67.5 | $54.4 \pm 1.0$ | $90.6 \pm 0.3$ |
| ERM | $90.2 \pm 0.3$ | 85.5 | $54.4 \pm 1.0$ | $90.6 \pm 0.3$ | $59.2 \pm 1.1$ | $90.3 \pm 0.1$ |
| MLDG | $92.2 \pm 0.1$ | 85.9 | $80.9 \pm 0.3$ | $70.6 \pm 0.2$ | $53.6 \pm 2.1$ | $91.5 \pm 1.1$ |
| CIDA | $92.0 \pm 1.2$ | $85.1^*$ | $85.2 \pm 1.4$ | $72.1 \pm 1.2$ | $50.5 \pm 1.5$ | $92.3 \pm 0.4$ |
| EAML | $92.2 \pm 0.5$ | $82.0^*$ | $84.7 \pm 0.4$ | $71.5 \pm 0.4$ | $61.0 \pm 2.8$ | $90.1 \pm 0.4$ |
| LSSAE | $92.5 \pm 0.4$ | $88.4^*$ | $85.5 \pm 0.3$ | $72.4 \pm 0.4$ | $88.4 \pm 0.8$ | $93.1 \pm 0.3$ |
| GI | $93.3 \pm 0.2$ | $83.2^*$ | $85.3 \pm 0.1$ | $71.8 \pm 0.2$ | $85.1 \pm 0.5$ | $93.7 \pm 0.2$ |
| DDA† | $93.3 \pm 1.2$ | $88.6 \pm 1.6$ | $85.0 \pm 1.1$ | $73.4 \pm 0.6$ | $91.2 \pm 3.3$ | $92.7 \pm 0.7$ |
| **DDA+TWA** | $\mathbf{94.0} \pm \mathbf{0.7}$ | $\mathbf{89.5} \pm \mathbf{0.5}$ | $\mathbf{86.0} \pm \mathbf{0.6}$ | $\mathbf{73.8} \pm \mathbf{1.2}$ | $\mathbf{92.7} \pm \mathbf{3.5}$ | $\mathbf{94.0} \pm \mathbf{1.2}$ |

$^*$ 15° results calculated from the Table 2 of Zeng et al. (2023). † Reproduced DDA results.

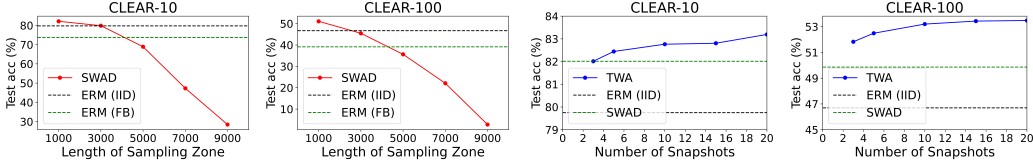

(a) SWAD (Cha et al., 2021) test acc (%) vs. sampling zone length

(b) TWA test acc (%) vs. snapshot number

Figure 3: Ablation results. (a) SWAD (Cha et al., 2021) performance drops significantly when using large sampling zone. Freezing BN layers also damage all performances with a bad basic model (ERM (FB)). (b) TWA performance increases with more snapshots, but the marginal growth decreases after 10 snapshots.

of improvement diminishes after 10 snapshots. Hence, we typically sample 10 snapshots to strike a balance between performance and computational cost.

## 5 CONCLUSION

In this paper we introduced Time-sensitive Weight Averaging (TWA) for Temporal Domain Generalization (TDG). Our TWA method uses a meta-learning approach that trains a Selector Network to predict coefficients from a timestamp for averaging a set of models into a time-sensitive network. We experiment on a wide set of real-world TGD benchmarks including CLEAR-10, CLEAR-100, Yearbook, and FMoW, where we report up to 4% boost in performance over state-of-the-art. In addition, we find that our approach can be combined with other methods, such as MIRO (Cha et al., 2022), to further boost performance. This demonstrates that our approach can generalize across many settings.

**Limitations.** A limitation of our work is that it is based on the assumption of smooth temporal shifts rather than abrupt ones, such as viral memes or unexpected natural disasters. Future work will explore how to enhance TDG for abrupt shifts.

## 6 CODE OF ETHICS STATEMENT

Like many works in AI, ours could be used for both benefit and harm. For example, one could use TWA to monitor and mitigate climate change or the spread of diseases. On the other hand, our contributions might help a bad actor build a stronger surveillance system, or spread misinformation. It is our responsibility as researchers, and as a society, to ensure that techniques like ours are used for good.

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

# A  APPENDIX

Our supplementary is organized as follows. Sec. B gives further details on our implementation and hyperparameter settings. Sec. C presents additional experiments, including an alternative method of creating expert snapshots, experiments over NLP benchmarks, efficiency evaluation of TWA, detailed results on each target domain, and comparison with DRAIN(ying Bai et al., 2022). Sec. D gives a full proof Theorem 1 from the main paper, as well as a refinement of it and an alternative bound that may be tighter in some special cases. Sec. E presents a toy experiment that illustrates the benefits of time-sensitive weight-averaging and looks at the role of temporal complexity. Sec. F provides pseudocodes for the training and inference algorithms of TWA.

# B  IMPLEMENTATION DETAILS

## B.1  CLEAR-10 AND CLEAR-100

**Dataset Settings** For CLEAR-10 and CLEAR-100, we set the image input shape as $(224, 224, 3)$ and using random crops and horizontal flips as data augmentations for all methods. We set $I_\theta$, the number of iterations to optimize $\theta$, to 70 epochs over the $s$ source domains $D_1, ..., D_s$ for different timestamp splits $t_s$.

**SWAD & MIRO** For SWAD and MIRO on CLEAR-10 and CLEAR-100, we used $20\%$ of the samples from source domains as the validation set and used the rest for training. We set the sampling zone length to be 1500 iterations. The batch size is set to be $8 \cdot s$, where $s$ is the number of source domains. We used Adam optimizer with a learning rate of $3 \times 10^{-4}$ for training from scratch, and $1 \times 10^{-4}$ for training with CB0 pre-training. We use the default values from the original authors for other hyperparameters.

**TWA** When applying TWA on CLEAR-10 and CLEAR-100, we use a 2-layer MLP as the selector network. The output dimension of Time2Vec that is concatented to the image features is set to be 128, and the hidden feature dimension of the MLP is set to be 64. When optimizing $\theta$, we used the same settings as those of SWAD and MIRO. When optimizing $\phi$, we typically use 2000 extra iterations. We also used Adam optimizer for $\phi$ with learning rate set to $3 \times 10^{-3}$, and the batch size is set to be 32 with gradient accumulation.

**CIDA & PCIDA** When applying CIDA and PCIDA on CLEAR-10 and CLEAR-100, we concatenate the timestamp to the $4^{th}$ dimmension of the network, and, thus, the ResNet-18 input size becomes $(224, 224, 4)$, without changing other network structures. The ratio of adversarial loss is set to be 2.0. We used Adam optimizer, and used exponential learning rate decay with the initial learning rate to be $1 \times 10^{-3}$. The batch size is set to be 64. Other hyperparameters are kept as default.

**AdaGraph** When applying AdaGraph to CLEAR-10 and CLEAR-100, we used 70 epochs for training the model on source domains, with 10 extra epochs to train the BatchNorm parameter estimation. We also used Adam optimizer. The initial learning rate is also $1 \times 10^{-3}$ and we also use exponential learning rate decay. The batch size is also set to be 64.

## B.2  YEARBOOK

**Dataset Settings** On Yearbook, we set the image input shape as $(32, 32, 1)$ with no extra data augmentation for all methods. And we set the $I_\theta$ equal to 20 epochs over the source domains $D_1, ..., D_s$ for different split timestamps $t_s$.

**SWAD** For SWAD on Yearbook, we use learning rate $1 \times 10^{-4}$ and sampling zone length 1000. The batch size is $\lfloor 0.8 \times s \rfloor$. Other settings are the same as those for CLEAR.

**TWA** When applying TWA to Yearbook, we only use 1000 extra iterations for optimizing the selector network. Batch size for optimizing $\theta$ is $\lfloor 0.8 \times s \rfloor$. Other settings are the same as those for CLEAR.

**CIDA & PCIDA** When applying CIDA and PCIDA on Yearbook, we concatenate the timestamp to the $2^{nd}$ dimension of the network input, and thus the ResNet-18 input size becomes $(32, 32, 2)$,

Table 8: CLEAR-10 test accuracies $(\%)$ on target domains, using ERM, AdaGraph (Mancini et al., 2019), CIDA/PCIDA (Wang et al., 2020), SWAD (Cha et al., 2021) and our TWA.

| Model | Method | $D_{10}$ | $D_{9-10}$ | $D_{8-10}$ | $D_{6-10}$ | $D_{4-10}$ |
|---|---|---|---|---|---|---|
| ResNet-18 | ERM (IID) | $86.7 \pm 0.6$ | $83.8 \pm 0.5$ | $84.3 \pm 1.5$ | $80.6 \pm 1.4$ | $76.3 \pm 0.9$ |
| | ERM (Last) | $86.3 \pm 2.0$ | $84.3 \pm 0.9$ | $82.5 \pm 1.5$ | $79.4 \pm 1.1$ | $76.6 \pm 0.8$ |
| | AdaGraph | $75,5 \pm 4.19$ | $74.8 \pm 1.9$ | $71.2 \pm 2.5$ | $56.4 \pm 3.5$ | $19.3 \pm 2.6$ |
| | CIDA | $85.6 \pm 0.5$ | $83.9 \pm 0.5$ | $81.3 \pm 0.9$ | $79.3 \pm 1.4$ | $71.5 \pm 4.3$ |
| | PCIDA | $87.3 \pm 0.8$ | $85.4 \pm 0.9$ | $82.9 \pm 1.0$ | $79.4 \pm 1.5$ | $72.4 \pm 2.2$ |
| | SWAD | $87.3 \pm 1.0$ | $86.0 \pm 0.6$ | $84.6 \pm 1.1$ | $81.5 \pm 1.4$ | $77.1 \pm 1.5$ |
| | **TWA-R** | $\mathbf{88.5} \pm \mathbf{0.3}$ | $\mathbf{87.3} \pm \mathbf{0.8}$ | $\mathbf{86.3} \pm \mathbf{0.2}$ | $\mathbf{83.0} \pm \mathbf{0.7}$ | $\mathbf{79.3} \pm \mathbf{0.3}$ |
| | **TWA-E** | $\mathbf{88.5} \pm \mathbf{0.2}$ | $\mathbf{87.3} \pm \mathbf{0.3}$ | $\mathbf{86.5} \pm \mathbf{0.3}$ | $82.8 \pm 0.4$ | $\mathbf{79.3} \pm \mathbf{0.4}$ |
| ResNet-50 | ERM (IID) | $85.9 \pm 1.6$ | $84.9 \pm 1.0$ | $84.7 \pm 0.2$ | $81.0 \pm 1.9$ | $77.1 \pm 0.6$ |
| | ERM (Last) | $85.6 \pm 1.4$ | $85.0 \pm 1.8$ | $84.5 \pm 0.4$ | $80.6 \pm 1.0$ | $76.0 \pm 1.5$ |
| | SWAD | $87.3 \pm 0.9$ | $86.5 \pm 0.2$ | $85.8 \pm 1.1$ | $82.2 \pm 1.6$ | $78.5 \pm 1.1$ |
| | **TWA-R** | $\mathbf{89.0} \pm \mathbf{0.3}$ | $\mathbf{87.8} \pm \mathbf{0.3}$ | $86.9 \pm 0.2$ | $\mathbf{84.0} \pm \mathbf{0.5}$ | $\mathbf{80.3} \pm \mathbf{0.4}$ |
| | **TWA-E** | $88.8 \pm 0.5$ | $87.1 \pm 0.3$ | $\mathbf{87.1} \pm \mathbf{0.1}$ | $\mathbf{84.0} \pm \mathbf{0.4}$ | $\mathbf{80.3} \pm \mathbf{0.2}$ |

without changing other network structures. Other settings are kept the same as those of CLEAR. Other settings are the same as those for CLEAR.

### B.3 FMoW-Time

**Dataset Settings** We set the image input shape as $(224, 224, 3)$ with normalizing the image only. We set $I_\theta$ to 30 epochs over source domains $D_1, ..., Ds$ for different split timestamps $t_s$.

**SWAD & MIRO & TWA** We used $3 \times 10^{-4}$ as the learning rate with pre-training on FMoW-Time. All the other settings are the same as those for CLEAR.

## C ADDITIONAL EXPERIMENTAL RESULTS

### C.1 EXPERT SNAPSHOTS

Our TWA method uses random sampling with late sampling strategy to get snapshots. But we have also tried using domain-specific fine-tuning to obtain snapshots, which we refer to as "Expert Snapshots". First, we train a base model $\theta_{all}$ over all source domains, and separately fine-tune $s$ different models $\theta_1, ..., \theta_s$ on each of the $s$ source domains, using $\theta_{all}$ as pre-trained weights. In this approach, we avoid using late sampling by freezing BN layers when fine-tuning $\theta_1, ..., \theta_s$.

Expert Snapshot experiments were conducted on CLEAR-10 and CLEAR-100. We trained our models for 2000 iterations with a learning rate that is $\frac{1}{s}$ times the original learning rate. We use TWA-R to denote TWA with randomly sampled snapshots, which is the TWA method presented in the main paper, and we use TWA-E to denote TWA with Expert Snapshots. Results and comparisons are in Tab. 8 and Tab. 9. We can see that TWA-E can also produce strong performance, but cannot outperform TWA-R. We have chosen TWA-R as our primary approach due to its simplicity and performance, but we still present our exploration results on Expert Snapshots, for potential reference in future work.

### C.2 NATURAL LANGUAGE PROCESSING BENCHMARKS

In addition to the image benchmarks used in our paper, we have also evaluated our method on NLP benchmarks. For this purpose, we use the ArXiv and Huffpost benchmarks in Wild-Time.

**ArXiv** (Yao et al., 2022) The task is to predict the primary category of arXiv pre-prints given the paper title as input. The entire dataset includes 2,057,952 titles within 172 pre-print categories from 2007 - 2022.

**Huffpost** (Yao et al., 2022) The task is to identify tags of news articles from their headlines. The dataset includes 63,907 articles within 11 categories from 2012 - 2022 in total.

Table 9: CLEAR-100 test accuracies (%) on target domains using ERM, AdaGraph (Mancini et al., 2019), CIDA/PCIDA (Wang et al., 2020), SWAD (Cha et al., 2021) and our TWA.

| Model | Method | $D_{10}$ | $D_{9-10}$ | $D_{8-10}$ | $D_{6-10}$ | $D_{4-10}$ |
|---|---|---|---|---|---|---|
| ResNet-18 | ERM (IID) | $68.3 \pm 0.3$ | $66.9 \pm 0.7$ | $64.9 \pm 1.4$ | $60.4 \pm 0.5$ | $53.3 \pm 0.9$ |
| | ERM (Last) | $67.0 \pm 1.1$ | $66.3 \pm 0.6$ | $64.4 \pm 0.5$ | $60.2 \pm 0.9$ | $53.1 \pm 0.8$ |
| | AdaGraph | $50.2 \pm 4.5$ | $39.5 \pm 2.5$ | $35.0 \pm 2.3$ | $21.0 \pm 2.6$ | $5.1 \pm 0.8$ |
| | CIDA | $67.8 \pm 0.3$ | $66.8 \pm 1.0$ | $66.5 \pm 0.3$ | $61.4 \pm 0.9$ | $52.7 \pm 1.2$ |
| | PCIDA | $69.2 \pm 0.1$ | $67.8 \pm 0.7$ | $67.2 \pm 0.7$ | $61.3 \pm 1.2$ | $53.3 \pm 1.7$ |
| | SWAD | $69.4 \pm 0.7$ | $67.3 \pm 0.7$ | $65.6 \pm 1.0$ | $61.5 \pm 0.8$ | $53.7 \pm 1.0$ |
| | **TWA-R** | $\mathbf{72.1} \pm \mathbf{0.3}$ | $\mathbf{70.2} \pm \mathbf{0.3}$ | $\mathbf{68.9} \pm \mathbf{0.2}$ | $\mathbf{64.3} \pm \mathbf{0.1}$ | $\mathbf{57.4} \pm \mathbf{0.3}$ |
| | **TWA-E** | $\mathbf{71.2} \pm \mathbf{0.4}$ | $\mathbf{69.5} \pm \mathbf{0.3}$ | $\mathbf{68.0} \pm \mathbf{0.1}$ | $\mathbf{63.4} \pm \mathbf{0.2}$ | $\mathbf{56.8} \pm \mathbf{0.4}$ |
| ResNet-50 | ERM (IID) | $71.8 \pm 0.5$ | $69.1 \pm 0.4$ | $67.7 \pm 0.7$ | $63.5 \pm 0.3$ | $56.1 \pm 0.6$ |
| | ERM (Last) | $70.5 \pm 1.9$ | $68.5 \pm 1.3$ | $66.6 \pm 0.7$ | $63.3 \pm 1.3$ | $55.4 \pm 1.3$ |
| | SWAD | $72.1 \pm 0.5$ | $70.8 \pm 0.9$ | $69.0 \pm 0.9$ | $65.2 \pm 0.3$ | $57.8 \pm 1.1$ |
| | **TWA-R** | $\mathbf{75.1} \pm \mathbf{0.6}$ | $\mathbf{73.4} \pm \mathbf{0.4}$ | $\mathbf{72.0} \pm \mathbf{0.3}$ | $\mathbf{68.3} \pm \mathbf{0.1}$ | $\mathbf{60.4} \pm \mathbf{0.7}$ |
| | **TWA-E** | $\mathbf{74.3} \pm \mathbf{0.5}$ | $\mathbf{72.1} \pm \mathbf{0.5}$ | $\mathbf{70.6} \pm \mathbf{0.4}$ | $\mathbf{66.8} \pm \mathbf{0.6}$ | $\mathbf{59.6} \pm \mathbf{0.3}$ |

We optimize our model parameters $\theta$ with the default settings used in the original Wild-Time paper Yao et al. (2022), including the network architectures and the split timestamps. We show the results with different training iterations, while keeping the sampling zone length as 1000 iterations. The other settings for SWAD and TWA are the same as those of the image benchmarks.

Results are presented in the Tab. 10. We see that TWA continues to enhance model generalization even in language tasks.

Table 10: ArXiv and HuffPost (Yao et al., 2022) test accuracies (%) on target domains, using SWAD (Cha et al., 2021) and our TWA.

| Dataset | arXiv (Yao et al., 2022) | | | HuffPost (Yao et al., 2022) | | |
|---|---|---|---|---|---|---|
| Target Domains | $2020 - 2022$ | $2018 - 2022$ | $2016 - 2022$ | $2017 - 2018$ | $2016 - 2018$ | $2015 - 2018$ |
| ERM (IID) | $52.1 \pm 0.6$ | $49.4 \pm 0.2$ | $46.3 \pm 1.1$ | $74.0 \pm 0.5$ | $69.7 \pm 0.3$ | $68.1 \pm 1.2$ |
| ERM (Last) | $52.0 \pm 0.2$ | $49.0 \pm 0.1$ | $46.9 \pm 0.4$ | $73.7 \pm 0.2$ | $69.6 \pm 0.9$ | $66.7 \pm 1.1$ |
| SWAD | $\mathbf{52.9} \pm \mathbf{0.1}$ | $\mathbf{50.0} \pm \mathbf{0.1}$ | $\mathbf{47.5} \pm \mathbf{0.1}$ | $\mathbf{74.8} \pm \mathbf{0.1}$ | $\mathbf{70.6} \pm \mathbf{0.5}$ | $\mathbf{68.0} \pm \mathbf{0.6}$ |
| **TWA** | $52.7 \pm 0.1$ | $\mathbf{50.0} \pm \mathbf{0.1}$ | $47.4 \pm 0.1$ | $\mathbf{74.8} \pm \mathbf{0.3}$ | $\mathbf{70.6} \pm \mathbf{0.5}$ | $\mathbf{68.0} \pm \mathbf{0.7}$ |

## C.3 EFFICIENCY EVALUATION

TWA does introduce some additional computational cost for optimizing the selector. To comprehensively assess TWA's performance, we also provide the speed of TWA on various datasets and network structures in Tab. 11. We can observe that the additional TWA does not significantly increase training time, resulting in an increase of $2 - 20\%$ of the total cost.

Table 11: Time costs in GPU hours. TWA Cost is the cost of optimizing the selector. All time costs are evaluated on a Tesla V100 GPU. We evaluate the efficiency performances on CLEAR (Lin et al., 2022), Yearbook and FMoW-Time (Yao et al., 2022), with ResNets (He et al., 2016).

| Dataset | CLEAR-10 | | CLEAR-100 | | Yearbook | FMoW-Time |
|---|---|---|---|---|---|---|
| Model | ResNet-18 | ResNet-50 | ResNet-18 | ResNet-50 | ResNet-18 | ResNet-50 |
| TWA Cost | 0.2 | 0.4 | 0.2 | 0.4 | 0.1 | 0.6 |
| Total Cost | 1.9 | 3.1 | 8.4 | 10.6 | 0.5 | 4.8 |
| TWA Proportion | 10.5% | 12.9% | 2.4% | 3.8% | 20.0% | 12.5% |

## C.4 PERFORMANCES ON EACH TARGET DOMAIN

We train the model over source domains $D_{1-5}$ and evaluate the performances of ERM (Lats), SWAD and TWA on each single domain within $D_{6-10}$. The results are shown in Tab. 12. Generally, the performance decreases as the domain gap becomes larger. And TWA improves the performances on every target domain equally, without significant bias towards any specific domain.

Table 12: CLEAR test accuracies (%) on every single target domain within $D_{6-10}$ using SWAD (Cha et al., 2021) and TWA.

| Dataset | Model | Method | $D_6$ | $D_7$ | $D_8$ | $D_9$ | $D_{10}$ | $D_{6-10}$ |
|---|---|---|---|---|---|---|---|---|
| CLEAR-10 | R18 | ERM (Last) | 79.9 | 79.8 | 79.2 | 79.1 | 78.5 | 79.3 |
| | | SWAD | 81.3 | 80.6 | 81.7 | 81.5 | 82.0 | 81.4 |
| | | TWA | 82.6 | 82.4 | 83.2 | 82.9 | 83.8 | 83.0 |
| | R50 | ERM (Last) | 80.9 | 81.0 | 80.4 | 80.3 | 80.0 | 80.5 |
| | | SWAD | 82.3 | 81.4 | 82.6 | 81.8 | 82.6 | 82.1 |
| | | TWA | 84.1 | 83.4 | 84.5 | 83.7 | 84.5 | 84.0 |
| CLEAR-100 | R18 | ERM (Last) | 60.8 | 61.5 | 59.5 | 57.8 | 59.9 | 59.9 |
| | | SWAD | 61.0 | 60.9 | 60.8 | 60.3 | 60.5 | 60.7 |
| | | TWA | 64.1 | 64.1 | 64.0 | 63.6 | 63.2 | 63.8 |
| | R50 | ERM (Last) | 64.4 | 65.3 | 63.6 | 62.3 | 63.3 | 63.8 |
| | | SWAD | 65.9 | 66.0 | 65.2 | 64.3 | 64.6 | 65.2 |
| | | TWA | 68.4 | 68.2 | 67.9 | 67.7 | 67.3 | 67.9 |

## C.5 COMPARISON WITH DRAIN

As DRAIN(ying Bai et al., 2022) is hard to compare with different benchmarks due to its highly task-specific implementation details, we evaluate DDA(Zeng et al., 2023) and DDA + TWA on DRAIN's Rotated MNIST setting for comparison. We use the exactly same experimental settings as those used in DRAIN. The comparison results are in Tab. 13. We could see that, our TWA still better performance when compared with DRAIN. In addition, the comparison with GI in Tab. 7 can also complement the comparison with DRAIN, as GI is generally comparable to DRAIN in performance according to DRAIN(ying Bai et al., 2022).

Table 13: Comparison with Rotated MNIST of our TWA with ERM, CDOT (Ortiz-Jiménez et al., 2019), CIDA (Wang et al., 2020), GI (Nasery et al., 2021), DRAIN (ying Bai et al., 2022) and DDA (Zeng et al., 2023). The Acc is accuracy (in %). The experiment settings are the same as those of Rotated MNIST in Table 1 of DRAIN (ying Bai et al., 2022).

| Method | ERM | CDOT | CIDA | GI | DRAIN | DDA | DDA + TWA |
|---|---|---|---|---|---|---|---|
| Acc | $81.4 \pm 4.0$ | $85.8 \pm 1.0$ | $90.7 \pm 0.7$ | $92.3 \pm 1.3$ | $92.5 \pm 1.1$ | $91.3 \pm 1.5$ | $92.7 \pm 0.7$ |

## D THEORETICAL ANALYSIS OF TWA

This section walks through the theoretical analysis described in the main paper. It is organized as follows. Sec. D.1 gives complete definitions of all terms and assumptions, as well as some intermediate definitions that are used in the full proof. Sec. D.2 proves all necessary lemmas in full detail. These lemmas work by bounding future risk in terms of past risk, using the maximum possible difference in risk gap of any two selectors on the future vs the past; this difference is then shown to be approximated well by temporal complexity $\hat{d}_{\Phi\Gamma\Phi}$. Sec. D.3 proves Theorem 1 from these lemmas by bounding past risk in terms of empirical robust risk across the temporal domains. Sec. D.4 proves Theorem 2, which is a refinement of Theorem 1 that gives a tighter bound. Sec. D.5 proves Theorem 3, which gives an alternative bound that may be tighter when the selector hypothesis class is highly constrained.

## D.1 DEFINITIONS AND ASSUMPTIONS

### D.1.1 DATA DISTRIBUTION

Let $D_{\mathcal{P}}(x, t, y)$ and $D_{\mathcal{F}}(x, t, y)$ be joint distributions over $(x, t, y)$, denoting the true distributions of past and future data. $(x_1, t_1, y_1), ..., (x_m, t_m, y_m) \sim D_{\mathcal{P}}^m(x, t, y)$ are the training set. We assume that these distributions come from a mixture of $H$ stationary components $D(x, y|h)$, where only mixing probabilities $D(h|t)$ vary over time, and marginals for time are known and uniform. Specifically:

$$D_{\mathcal{P}}(x, t, y) := \sum_{h=1}^{H} D(x, y|h) D(h|t) D_{\mathcal{P}}(t) \qquad D_{\mathcal{F}}(x, t, y) := \sum_{h=1}^{H} D(x, y|h) D(h|t) D_{\mathcal{F}}(t) \quad (4)$$

$$D_{\mathcal{P}}(t) := \mathrm{Unif}([0, T_{\mathcal{P}}]) \qquad\qquad D_{\mathcal{F}}(t) := \mathrm{Unif}([T_{\mathcal{P}}, T_{\mathcal{P}} + T_{\mathcal{F}}]) \qquad (5)$$

where $T_{\mathcal{P}}$ is the past-vs-future "cutoff" time, and $T_{\mathcal{P}} + T_{\mathcal{F}}$ is the end of the future. Note that $D(h|t)$ is the source of all temporal domain shift.

We assume that the *speed* of domain-shift is bounded, and that the rate of change of the *velocity* of this shift is also bounded. Specifically, $\exists$ constants $\eta$, $\rho$, and $\tau$ such that, for all times $t$ and $t^{'} > t$:

$$\left\| D(h|t^{'}) - D(h|t) \right\|_1 \le \eta(t^{'} - t) \qquad (6)$$

$$\left\| \frac{D(h|t^{'}) - D(h|t)}{t^{'} - t} - \frac{D(h|t) - D(h|t - \tau)}{\tau} \right\|_1 \le \rho(t^{'} - t) \qquad (7)$$

Note that if we multiply both sides of the second inequality by $t^{'} - t$, we see that it implies that a linear extrapolation of $D(h|t^{'})$ from $D(h|t)$ and $D(h|t - \tau)$ will be off by at most $\rho(t^{'} - t)^2$. We will use such an extrapolation in our temporal complexity term $\hat{d}_{\Phi\Gamma\Phi}$.

### D.1.2 LOSS AND RISK TERMS

Our risks all depend on the loss function $l(\cdot, \cdot)$, which is the 0-1 loss.

We start by defining **distributional risks**:

- $\zeta^{\theta_{1:K}}(\mu, t) := E_{(x,y)\sim D(x,y|t)}[l((\theta_{1:K} * \mu)(x), y)]$ is the distributional risk caused by weight-averaging coefficients $\mu$ on $(x, y)$ pairs sampled at time $t$. Note that $t$ could be in the future or the past for this definition.
- $\mathcal{E}_{D_{\mathcal{P}}}(\phi) := E_{t\sim D_{\mathcal{P}}(t)}[\zeta^{\theta_{1:K}}(\phi(t), t)]$ is distributional risk caused by selector $\phi$ on past data.
- $\mathcal{E}_{D_{\mathcal{F}}}(\phi) := E_{t\sim D_{\mathcal{F}}(t)}[\zeta^{\theta_{1:K}}(\phi(t), t)]$ is distributional risk on future data.
- $\mathcal{E}_{D_{\mathcal{P} \bigcup \mathcal{F}}}(\phi) := E_{t\sim D_{\mathcal{P} \bigcup \mathcal{F}}(t)}[\zeta^{\theta_{1:K}}(\phi(t), t)]$ is distributional risk on data from $D_{\mathcal{P} \bigcup \mathcal{F}}$, which is an equal mixture of past and future data.
- $G : \Delta_K \to [0, 1]^H$ is a function such that $G(\mu)_h := E_{(x,y)\sim D(x,y|h)}[l((\theta_{1:K} * \mu)(x), y)]$. In other words, $G(\mu)_h$ tells us the risk that weight-averaging coefficients $\mu$ would cause on data from the $h$-th mixture component. Note that $\zeta^{\theta_{1:K}}(\mu, t) = G(\mu) \cdot D(h|t)$.

We also define a **best-in-family selector and an oracle risk**:

- $\phi^*_{D_{\mathcal{P} \bigcup \mathcal{F}}} := \mathrm{argmin}_{\phi\in\Phi} \mathcal{E}_{D_{\mathcal{P} \bigcup \mathcal{F}}}(\phi)$ is the "best-in-family" selector for data from both past and future. It depends on the capacity of selector hypothesis class $\Phi$.
- $\lambda^*_{D_{\mathcal{P}}} := E_{t\sim D_{\mathcal{P}}(t)}[\inf_{\mu\in\Delta_K} \zeta^{\theta_{1:K}}(\mu, t)]$ is the risk that would be incurred by an "oracle" that could independently choose a $\mu$ for each $t$. It is not limited by $\Phi$, but only by snapshots $\theta_{1:K}$ and the Bayes risk of the data distribution itself.

Our proof uses the concept of **"risk-gaps"**, which we define below:

- $\mathcal{E}_{D_{\mathcal{P}}}(\phi^{'}, \phi^{''}) := E_{t\sim D_{\mathcal{P}}(t)}[|\zeta^{\theta_{1:K}}(\phi^{'}(t), t) - \zeta^{\theta_{1:K}}(\phi^{''}(t), t)|]$

- $\mathcal{E}_{D_{\mathcal{F}}}(\phi^{'}, \phi^{''}) := E_{t \sim D_{\mathcal{F}}(t)}[|\zeta^{\theta_{1:K}}(\phi^{'}(t), t) - \zeta^{\theta_{1:K}}(\phi^{''}(t), t)|]$

Our proof relies on the technique of discretizing the past data into **"time-bins"**, so we give some definitions related to that:

- $B(t) := [(\lceil t / \frac{T_{\mathcal{P}}}{\sqrt{m}} \rceil - 1) \frac{T_{\mathcal{P}}}{\sqrt{m}}, \lceil t / \frac{T_{\mathcal{P}}}{\sqrt{m}} \rceil \frac{T_{\mathcal{P}}}{\sqrt{m}}]$ maps time $t$ to the "time-bin" that contains it. Note that each bin has width $T_{\mathcal{P}} / \sqrt{m}$, so there are $\sqrt{m}$ bins total.
- $\xi^t(\mu) := E_{s \sim \text{Unif}(B(t))}[\zeta^{\theta_{1:K}}(\mu, s)]$ is the risk caused by coefficients $\mu$ on data whose time is in the same bin as $t$. Note that $\xi^t(\mu)$ is the same for all values of $t$ within the same bin.
- $\overline{D}^t(h) := E_{s \sim \text{Unif}(B(t))}[D(h|s)]$ is shorthand denoting expectation of mixing probabilities over time-bin containing $t$. Again, this is the same for all values of $t$ within the same bin.

Finally, we define some **empirical risks**.

- $\hat{\xi}^t(\mu) := \frac{1}{\sqrt{m}} \sum_{i:t_i \in B(t)} l((\theta_{1:K} * \mu)(x_i), y_i)$ is the empirical risk caused by coefficients $\mu$ on training data from bin $B(t)$. Note that we assume there are exactly $\sqrt{m}$ training points in each of the $\sqrt{m}$ bins. We use this definition in our temporal complexity measure.
- $\hat{\xi}^{\gamma,t}(\mu) := \max_{\|\beta\| \le \gamma} \frac{1}{\sqrt{m}} \sum_{i:t_i \in B(t)} l((\theta_{1:K} * \mu + \beta)(x_i), y_i)$ is the *robust* version of the above. We use this definition in Theorems 1 and 2.
- $\hat{\mathcal{E}}_{D_{\mathcal{P}}}(\phi) := \frac{1}{m} \sum_{i=1}^{m} l((\theta_{1:K} * \phi(t_i))(x_i), y_i)$ is the plain empirical risk of selector $\phi$ on the entire training set. We use this definition in Theorem 3.

### D.1.3 TEMPORAL COMPLEXITY

Inspired by the $d_{\mathcal{H} \Delta \mathcal{H}}$ divergence from Ben-David et al. (2010), we define a temporal complexity metric that only depends on selector hypothesis class $\Phi$ and training dataset. Our metric is:

$$\hat{d}_{\Phi\Gamma\Phi} := \sup_{\phi^{'}, \phi^{''} \in \Phi} E_{t \sim D_{\mathcal{F}}(t)}[|\hat{\aleph}_{\mathcal{F}}(\phi^{'}, t) - \hat{\aleph}_{\mathcal{F}}(\phi^{''}, t)|] - E_{t \sim D_{\mathcal{P}}(t)}[|\hat{\aleph}_{\mathcal{P}}(\phi^{'}, t) - \hat{\aleph}_{\mathcal{P}}(\phi^{''}, t)|] \quad (8)$$

where $\hat{\aleph}_{\mathcal{F}}(\phi, t) := \hat{\xi}^{T_{\mathcal{P}}}(\phi(t)) + \frac{t - T_{\mathcal{P}}}{\tau}(\hat{\xi}^{T_{\mathcal{P}}}(\phi(t)) - \hat{\xi}^{T_{\mathcal{P}} - \tau}(\phi(t)))$ and $\hat{\aleph}_{\mathcal{P}}(\phi, t) := \hat{\xi}^t(\phi(t))$ and $\hat{\xi}^t(\mu) := \frac{1}{\sqrt{m}} \sum_{i:t_i \in B(t)} l((\theta_{1:K} * \mu)(x_i), y_i)$.

$\hat{d}_{\Phi\Gamma\Phi}$ estimates how much the risk gap between any two selectors can grow from past to future. Note that $\hat{\aleph}_{\mathcal{F}}(\phi, t)$ is in some sense a linear extrapolation of future risk from past risk, except that it feeds future timestamps into the selector net before applying its outputted coefficients to past training data.

### D.2 LEMMAS

We begin our proof with similar reasoning to Ben-David et al. (2010) to derive a bound on $\mathcal{E}_{D_{\mathcal{F}}}(\phi)$ in terms of $\mathcal{E}_{D_{\mathcal{P}}}(\phi)$ using the "risk-gap" concept defined earlier.

**Lemma 1.** *The following bound holds for all $\phi \in \Phi$:*

$$\mathcal{E}_{D_{\mathcal{F}}}(\phi) \le \mathcal{E}_{D_{\mathcal{P}}}(\phi) + 2(\mathcal{E}_{D_{\mathcal{P} \bigcup \mathcal{F}}}(\phi^*_{D_{\mathcal{P} \bigcup \mathcal{F}}}) - \lambda^*_{D_{\mathcal{P}}}) + \sup_{\phi^{'}, \phi^{''} \in \Phi} (\mathcal{E}_{D_{\mathcal{F}}}(\phi^{'}, \phi^{''}) - \mathcal{E}_{D_{\mathcal{P}}}(\phi^{'}, \phi^{''})) \quad (9)$$

*Proof.* We start by noting the following fact about any three non-negative numbers $a, b \ge c \ge 0$:

$$|a - b| = |(a - c) - (b - c)| \le |a - c| + |b - c| = (a - c) + (b - c) = a + b - 2c \quad (10)$$

Substituting $\zeta^{\theta_{1:K}}(\phi(t), t)$, $\zeta^{\theta_{1:K}}(\phi^*_{D_{\mathcal{P} \bigcup \mathcal{F}}}(t), t)$, and $\inf_{\mu \in \Delta_K} \zeta^{\theta_{1:K}}(\mu, t)$ for $a$, $b$, and $c$, respectively, and taking the expectation over $t \sim D_{\mathcal{P}}(t)$ gets us the following useful fact about $\mathcal{E}_{D_{\mathcal{P}}}(\phi, \phi^*_{D_{\mathcal{P} \bigcup \mathcal{F}}})$:

$$\mathcal{E}_{D_{\mathcal{P}}}(\phi, \phi^*_{D_{\mathcal{P} \bigcup \mathcal{F}}}) \le \mathcal{E}_{D_{\mathcal{P}}}(\phi) + \mathcal{E}_{D_{\mathcal{P}}}(\phi^*_{D_{\mathcal{P} \bigcup \mathcal{F}}}) - 2\lambda^*_{D_{\mathcal{P}}} \tag{11}$$

We now use this fact in our derivation:

$$\mathcal{E}_{D_{\mathcal{F}}}(\phi) = E_{t \sim D_{\mathcal{F}}(t)}[\zeta^{\theta_{1:K}}(\phi(t), t)] \tag{12}$$

$$\le E_{t \sim D_{\mathcal{F}}(t)}[\zeta^{\theta_{1:K}}(\phi^*_{D_{\mathcal{P} \bigcup \mathcal{F}}}(t), t) + |\zeta^{\theta_{1:K}}(\phi(t), t) - \zeta^{\theta_{1:K}}(\phi^*_{D_{\mathcal{P} \bigcup \mathcal{F}}}(t), t)|] \tag{13}$$

$$= \mathcal{E}_{D_{\mathcal{F}}}(\phi^*_{D_{\mathcal{P} \bigcup \mathcal{F}}}) + \mathcal{E}_{D_{\mathcal{F}}}(\phi, \phi^*_{D_{\mathcal{P} \bigcup \mathcal{F}}}) \tag{14}$$

$$= \mathcal{E}_{D_{\mathcal{F}}}(\phi^*_{D_{\mathcal{P} \bigcup \mathcal{F}}}) + \mathcal{E}_{D_{\mathcal{P}}}(\phi, \phi^*_{D_{\mathcal{P} \bigcup \mathcal{F}}}) + (\mathcal{E}_{D_{\mathcal{F}}}(\phi, \phi^*_{D_{\mathcal{P} \bigcup \mathcal{F}}}) - \mathcal{E}_{D_{\mathcal{P}}}(\phi, \phi^*_{D_{\mathcal{P} \bigcup \mathcal{F}}})) \tag{15}$$

$$\le \mathcal{E}_{D_{\mathcal{P}}}(\phi) + 2(\mathcal{E}_{D_{\mathcal{P} \bigcup \mathcal{F}}}(\phi^*_{D_{\mathcal{P} \bigcup \mathcal{F}}}) - \lambda^*_{D_{\mathcal{P}}}) + (\mathcal{E}_{D_{\mathcal{F}}}(\phi, \phi^*_{D_{\mathcal{P} \bigcup \mathcal{F}}}) - \mathcal{E}_{D_{\mathcal{P}}}(\phi, \phi^*_{D_{\mathcal{P} \bigcup \mathcal{F}}})) \tag{16}$$

$$\le \mathcal{E}_{D_{\mathcal{P}}}(\phi) + 2(\mathcal{E}_{D_{\mathcal{P} \bigcup \mathcal{F}}}(\phi^*_{D_{\mathcal{P} \bigcup \mathcal{F}}}) - \lambda^*_{D_{\mathcal{P}}}) + \sup_{\phi', \phi'' \in \Phi}(\mathcal{E}_{D_{\mathcal{F}}}(\phi', \phi'') - \mathcal{E}_{D_{\mathcal{P}}}(\phi', \phi'')) \tag{17}$$

$$\square$$

Our next set of lemmas will lead to an upper bound on the supremum from the above lemma. We start by showing that $\hat{\xi}^t(\mu)$ is likely close to $\xi^t(\mu)$.

**Lemma 2.** *Fix $t$ to an arbitrary value in $[0, T_{\mathcal{P}}]$. Let $v_e$ be the VC-dimension of $CH(\theta_{1:K})$, where "CH" denotes a convex hull. With probability at least $1 - \frac{\delta}{2\sqrt{m}}$, the following holds for all $\mu \in \Delta_K$:*

$$|\hat{\xi}^t(\mu) - \xi^t(\mu)| \le \sqrt{\frac{4v_e(\ln(\sqrt{m}/v_e) + 1) + \ln(4\sqrt{m}/\delta)}{\sqrt{m}}} \tag{18}$$

*Proof.* We start by expanding both quantities by their definitions:

$$|\hat{\xi}^t(\mu) - \xi^t(\mu)| = |\frac{1}{\sqrt{m}} \sum_{i:t_i \in B(t)} l((\theta_{1:K} * \mu)(x_i), y_i) - E_{s \sim \text{Unif}(B(t))}[\zeta^{\theta_{1:K}}(\mu, s)]| \tag{19}$$

$$= |\frac{1}{\sqrt{m}} \sum_{i:t_i \in B(t)} l((\theta_{1:K} * \mu)(x_i), y_i) - E_{s \sim \text{Unif}(B(t)), (x,y) \sim D(x,y|s)}[l((\theta_{1:K} * \mu)(x), y)]| \tag{20}$$

We note that $\tilde{\theta} := \theta_{1:K} * \mu \in CH(\theta_{1:K})$ for all $\mu \in \Delta_K$, and so the following is true:

$$\begin{aligned} &\sup_{\mu \in \Delta_K} |\hat{\xi}^t(\mu) - \xi^t(\mu)| \\ &\le \sup_{\tilde{\theta} \in CH(\theta_{1:K})} |\frac{1}{\sqrt{m}} \sum_{i:t_i \in B(t)} l(\tilde{\theta}(x_i), y_i) - E_{s \sim \text{Unif}(B(t)), (x,y) \sim D(x,y|s)}[l(\tilde{\theta}(x), y)]| \end{aligned} \tag{21}$$

This is a supremum over the difference in empirical and distributional risk of $\tilde{\theta} \in CH(\theta_{1:K})$, where the distribution is over $(x, y)$ pairs from time-bin $B(t)$ and the number of empirical datapoints is $\sqrt{m}$. Thus, we can bound this supremum with a standard VC-bound which is $\sqrt{\frac{4v_e(\ln(\sqrt{m}/v_e) + 1) + \ln(4\sqrt{m}/\delta)}{\sqrt{m}}}$, which concludes the proof of this lemma. $\square$

We next use the "first-derivative" bound on $D(h|t)$ to show that $\xi^t(\mu)$ is close to $\zeta^{\theta_{1:K}}(\mu, t)$.

**Lemma 3.** *The following bound holds for all $t \in [0, T_{\mathcal{P}}]$ and all $\mu \in \Delta_K$:*

$$|\xi^t(\mu) - \zeta^{\theta_{1:K}}(\mu, t)| \le \eta \frac{T_{\mathcal{P}}}{\sqrt{m}} \tag{22}$$

*Proof.* We start by expanding $\xi^t(\mu)$ and noting that $\zeta^{\theta_{1:K}}(\mu, t) = G(\mu) \cdot D(h|t)$ to get:

$$|\xi^t(\mu) - \zeta^{\theta_{1:K}}(\mu, t)| = |E_{s \sim \text{Unif}(B(t))}[\zeta^{\theta_{1:K}}(\mu, s)] - \zeta^{\theta_{1:K}}(\mu, t)| \tag{23}$$

$$= |E_{s \sim \text{Unif}(B(t))}[G(\mu) \cdot D(h|s)] - G(\mu) \cdot D(h|t)| \tag{24}$$

$$= |G(\mu) \cdot (E_{s \sim \text{Unif}(B(t))}[D(h|s)] - D(h|t))| \tag{25}$$

$$= |G(\mu) \cdot (\overline{D}^t(h) - D(h|t))| \tag{26}$$

$$\leq \left\| \overline{D}^t(h) - D(h|t) \right\|_1 \tag{27}$$

where the last line is true because $G(\mu) \in [0, 1]^H$.

Now, we note that $\overline{D}^t(h), D(h|t) \in \text{CH}(\{D(h|s) : s \in B(t)\})$. Therefore:

$$\left\| \overline{D}^t(h) - D(h|t) \right\|_1 \leq \text{diam}(\text{CH}(\{D(h|s) : s \in B(t)\})) \tag{28}$$

$$= \text{diam}(\{D(h|s) : s \in B(t)\}) \tag{29}$$

$$= \sup_{s', s'' \in B(t)} \left\| D(h|s'') - D(h|s') \right\|_1 \tag{30}$$

$$\leq \eta \frac{T_{\mathcal{P}}}{\sqrt{m}} \tag{31}$$

where the last line is due to the first bound in 6 and the fact that a time-bin has width $\frac{T_{\mathcal{P}}}{\sqrt{m}}$. $\square$

Our next lemma combines the previous two lemmas to show that $\hat{\xi}^t(\mu)$ is close to $\zeta^{\theta_{1:K}}(\mu, t)$.

**Lemma 4.** *Fix $t$ to an arbitrary value in $[0, T_{\mathcal{P}}]$. Let $v_e$ be the VC-dimension of $CH(\theta_{1:K})$. With probability at least $1 - \frac{\delta}{2\sqrt{m}}$, the following bound holds for all $\mu \in \Delta_K$:*

$$|\hat{\xi}^t(\mu) - \zeta^{\theta_{1:K}}(\mu, t)| \leq \eta \frac{T_{\mathcal{P}}}{\sqrt{m}} + \sqrt{\frac{4v_e(\ln(\sqrt{m}/v_e)+1) + \ln(4\sqrt{m}/\delta)}{\sqrt{m}}} \tag{32}$$

*Proof.* This follows immediately from combining Lemma 2 with Lemma 3. $\square$

We will next show that $\hat{\aleph}_{\mathcal{F}}(\phi, t)$ is likely close to $\zeta^{\theta_{1:K}}(\phi(t), t)$ for any *future* $t$.

**Lemma 5.** *Fix $t$ to an arbitrary value in $[T_{\mathcal{P}}, T_{\mathcal{P}} + T_{\mathcal{F}}]$. Let $v_e$ be the VC-dimension of $CH(\theta_{1:K})$. With probability at least $1 - \frac{\delta}{2}$, the following bound holds for all $\phi \in \Phi$:*

$$|\hat{\aleph}_{\mathcal{F}}(\phi, t) - \zeta^{\theta_{1:K}}(\phi(t), t)|$$
$$\leq (1 + 2\frac{t - T_{\mathcal{P}}}{\tau})(\eta \frac{T_{\mathcal{P}}}{\sqrt{m}} + \sqrt{\frac{4v_e(\ln(\sqrt{m}/v_e)+1) + \ln(4\sqrt{m}/\delta)}{\sqrt{m}}}) + \rho(t - T_{\mathcal{P}})^2 \tag{33}$$

*Proof.* We start by expanding the definition of $\hat{\aleph}_{\mathcal{F}}$ (8) and combining with Lemma 4 to conclude that, with probability at least $1 - \frac{\delta}{2}$, the following holds for all $\phi \in \Phi$:

$$|\hat{\aleph}_{\mathcal{F}}(\phi, t) - (\zeta^{\theta_{1:K}}(\phi(t), T_{\mathcal{P}}) + \frac{t - T_{\mathcal{P}}}{\tau}(\zeta^{\theta_{1:K}}(\phi(t), T_{\mathcal{P}}) - \zeta^{\theta_{1:K}}(\phi(t), T_{\mathcal{P}} - \tau)))| \tag{34}$$

$$\leq (1 + 2\frac{t - T_{\mathcal{P}}}{\tau})(\eta \frac{T_{\mathcal{P}}}{\sqrt{m}} + \sqrt{\frac{4v_e(\ln(\sqrt{m}/v_e)+1) + \ln(4\sqrt{m}/\delta)}{\sqrt{m}}}) \tag{35}$$

The "all $\phi \in \Phi$" part is valid because $\{\phi(t) : \phi \in \Phi\} \subseteq \Delta_K$, and the "$1 - \frac{\delta}{2}$" part is valid because there are at most $\sqrt{m}$ possible failure events (one for each time-bin), each with failure probability at most $\frac{\delta}{2\sqrt{m}}$, so a union bound can be applied.

Next, we use a linear extrapolation:

$$|\zeta^{\theta_{1:K}}(\phi(t), t) - (\zeta^{\theta_{1:K}}(\phi(t), T_\mathcal{P}) + \frac{t - T_\mathcal{P}}{\tau}(\zeta^{\theta_{1:K}}(\phi(t), T_\mathcal{P}) - \zeta^{\theta_{1:K}}(\phi(t), T_\mathcal{P} - \tau)))| \quad (36)$$

$$= |G(\phi(t)) \cdot (D(h|t) - (D(h|T_\mathcal{P}) + \frac{t - T_\mathcal{P}}{\tau}(D(h|T_\mathcal{P}) - D(h|T_\mathcal{P} - \tau))))| \quad (37)$$

$$\leq \left\| D(h|t) - (D(h|T_\mathcal{P}) + \frac{t - T_\mathcal{P}}{\tau}(D(h|T_\mathcal{P}) - D(h|T_\mathcal{P} - \tau))) \right\|_1 \quad (38)$$

$$= (t - T_\mathcal{P}) \left\| \frac{D(h|t) - D(h|T_\mathcal{P})}{t - T_\mathcal{P}} - \frac{D(h|T_\mathcal{P}) - D(h|T_\mathcal{P} - \tau)}{\tau} \right\|_1 \quad (39)$$

$$\leq \rho(t - T_\mathcal{P})^2 \quad (40)$$

Third-last line is valid because $G(\phi(t)) \in [0, 1]^H$, and last line is valid due to second bound in 6.

Putting these two parts together gets us the conclusion of the lemma. $\qquad \square$

We are now ready to prove a bound on the supremum from Lemma 1.

**Lemma 6.** *Let $v_e$ be VC-dimension of $CH(\theta_{1:K})$. With probability at least $1 - \frac{\delta}{2}$, the following holds:*

$$\sup_{\phi', \phi'' \in \Phi} (\mathcal{E}_{D_\mathcal{F}}(\phi', \phi'') - \mathcal{E}_{D_\mathcal{P}}(\phi', \phi''))$$

$$\leq \hat{d}_{\Phi\Gamma\Phi} + 2(2 + \frac{T_\mathcal{F}}{\tau})(\eta\frac{T_\mathcal{P}}{\sqrt{m}} + \sqrt{\frac{4v_e(\ln(\sqrt{m}/v_e) + 1) + \ln(4\sqrt{m}/\delta)}{\sqrt{m}}}) + \frac{2}{3}\rho T_\mathcal{F}^2 \quad (41)$$

*Proof.* First, we use the definitions of $\mathcal{E}_{D_\mathcal{P}}(\phi', \phi'')$ and $\hat{\aleph}_\mathcal{P}$, as well as Lemma 4, and take expectation over $t$ to conclude that, with probability at least $1 - \frac{\delta}{2}$, the following is true for all $\phi', \phi'' \in \Phi$:

$$|E_{t \sim D_\mathcal{P}(t)}[|\hat{\aleph}_\mathcal{P}(\phi', t) - \hat{\aleph}_\mathcal{P}(\phi'', t)|] - \mathcal{E}_{D_\mathcal{P}}(\phi', \phi'')| \quad (42)$$

$$\leq E_{t \sim D_\mathcal{P}(t)}[||\hat{\xi}^t(\phi'(t)) - \hat{\xi}^t(\phi''(t))| - |\zeta^{\theta_{1:K}}(\phi'(t), t) - \zeta^{\theta_{1:K}}(\phi''(t), t)||] \quad (43)$$

$$\leq 2(\eta\frac{T_\mathcal{P}}{\sqrt{m}} + \sqrt{\frac{4v_e(\ln(\sqrt{m}/v_e) + 1) + \ln(4\sqrt{m}/\delta)}{\sqrt{m}}}) \quad (44)$$

Next, we use the definition of $\mathcal{E}_{D_\mathcal{F}}(\phi', \phi'')$, as well as Lemma 5, and take expectation over $t$ to conclude that, with probability at least $1 - \frac{\delta}{2}$, the following is true for all $\phi', \phi'' \in \Phi$:

$$|E_{t \sim D_\mathcal{F}(t)}[|\hat{\aleph}_\mathcal{F}(\phi', t) - \hat{\aleph}_\mathcal{F}(\phi'', t)|] - \mathcal{E}_{D_\mathcal{F}}(\phi', \phi'')| \quad (45)$$

$$\leq E_{t \sim D_\mathcal{F}(t)}[||\hat{\aleph}_\mathcal{F}(\phi', t) - \hat{\aleph}_\mathcal{F}(\phi'', t)| - |\zeta^{\theta_{1:K}}(\phi'(t), t) - \zeta^{\theta_{1:K}}(\phi''(t), t)||] \quad (46)$$

$$\leq E_{t \sim D_\mathcal{F}(t)}[2(1 + 2\frac{t - T_\mathcal{P}}{\tau})(\eta\frac{T_\mathcal{P}}{\sqrt{m}} + \sqrt{\frac{4v_e(\ln(\sqrt{m}/v_e) + 1) + \ln(4\sqrt{m}/\delta)}{\sqrt{m}}}) + \rho(t - T_\mathcal{P})^2] \quad (47)$$

$$= 2(1 + \frac{T_\mathcal{F}}{\tau})(\eta\frac{T_\mathcal{P}}{\sqrt{m}} + \sqrt{\frac{4v_e(\ln(\sqrt{m}/v_e) + 1) + \ln(4\sqrt{m}/\delta)}{\sqrt{m}}}) + \frac{2}{3}\rho T_\mathcal{F}^2 \quad (48)$$

Putting these two bounds together and taking the supremum over $\phi^{'}, \phi^{''} \in \Phi$ gets us the conclusion of the lemma. Note that the failure probability is at most $\frac{\delta}{2}$ because there are at most $\sqrt{m}$ possible failure events, one per time-bin, each with failure probability at most $\frac{\delta}{2\sqrt{m}}$. $\qquad\square$

We are now ready to bound $\mathcal{E}_{D_{\mathcal{F}}}(\phi)$ in terms of $\mathcal{E}_{D_{\mathcal{P}}}(\phi)$ and $\hat{d}_{\Phi\Gamma\Phi}$.

**Lemma 7.** *Let $v_e$ be the VC-dimension of $CH(\theta_{1:K})$. With probability at least $1 - \frac{\delta}{2}$, the following bound holds for all $\phi \in \Phi$:*

$$
\mathcal{E}_{D_{\mathcal{F}}}(\phi) \leq \mathcal{E}_{D_{\mathcal{P}}}(\phi) + 2(\mathcal{E}_{D_{\mathcal{P}\bigcup\mathcal{F}}}(\phi^*_{D_{\mathcal{P}\bigcup\mathcal{F}}}) - \lambda^*_{D_{\mathcal{P}}})
$$
$$
+ \hat{d}_{\Phi\Gamma\Phi} + 2(2 + \frac{T_{\mathcal{F}}}{\tau})(\eta\frac{T_{\mathcal{P}}}{\sqrt{m}} + \sqrt{\frac{4v_e(\ln(\sqrt{m}/v_e)+1) + \ln(4\sqrt{m}/\delta)}{\sqrt{m}}}) + \frac{2}{3}\rho T_{\mathcal{F}}^2 \tag{49}
$$

*Proof.* This follows immediately from combining Lemma 6 with Lemma 1. $\qquad\square$

### D.3 RESTATEMENT AND PROOF OF THEOREM 1

We now restate Theorem 1 from the main paper, and prove it using the above lemmas.

**Theorem 1.** *Let $\Theta_1, ..., \Theta_N$ be a finite cover of model space $\Theta \subset \mathbb{R}^d$, where $N := \lceil(diam(\Theta)/\gamma)^d\rceil$. Let $v_j$ be the VC-dimension of $\Theta_j$, and let $v_e$ be the VC-dimension of $CH(\theta_{1:K})$, which is the convex hull of the snapshots. Then, with probability at least $1 - \delta$, the following bound holds for all $\phi \in \Phi$:*

$$
\mathcal{E}_{D_{\mathcal{F}}}(\phi) \leq E_{t\sim D_{\mathcal{P}}(t)}[\hat{\xi}^{\gamma,t}(\phi(t))] + \max_j \sqrt{\frac{v_j(\ln(\sqrt{m}/v_j)+1) + \ln(4N\sqrt{m}/\delta)}{\sqrt{m}}}
$$
$$
+ 2(\mathcal{E}_{D_{\mathcal{P}\bigcup\mathcal{F}}}(\phi^*_{D_{\mathcal{P}\bigcup\mathcal{F}}}) - \lambda^*_{D_{\mathcal{P}}}) \tag{50}
$$
$$
+ \hat{d}_{\Phi\Gamma\Phi} + 2(\frac{5}{2} + \frac{T_{\mathcal{F}}}{\tau})(\eta\frac{T_{\mathcal{P}}}{\sqrt{m}} + \sqrt{\frac{4v_e(\ln(\sqrt{m}/v_e)+1) + \ln(4\sqrt{m}/\delta)}{\sqrt{m}}}) + \frac{2}{3}\rho T_{\mathcal{F}}^2
$$

*Proof.* First, we must prove an upper bound on $\mathcal{E}_{D_{\mathcal{P}}}(\phi)$ in terms of empirical risk. To derive this bound, we fix $t$ to an arbitrary value in $[0, T_{\mathcal{P}}]$ and consider the following fact:

$$
\sup_{\mu\in\Delta_K} |\hat{\xi}^{\gamma,t}(\mu) - \xi^t(\mu)|
$$
$$
\leq \sup_{\tilde{\theta}\in\Theta} |\max_{\|\beta\|\leq\gamma} \frac{1}{\sqrt{m}} \sum_{i:t_i\in B(t)} l((\tilde{\theta}+\beta)(x_i), y_i) - E_{s\sim\text{Unif}(B(t)),(x,y)\sim D(x,y|s)}[l(\tilde{\theta}(x), y)]| \tag{51}
$$

This is the supremum (over $\Theta$) of the absolute difference between empirical robust risk and distributional risk where the distribution is $(x, y)$ pairs from the $B(t)$ time-bin, and the number of empirical samples is $\sqrt{m}$. Therefore, we apply Lemma 2 from SWAD (Cha et al., 2021) to conclude that with probability at least $1 - \frac{\delta}{2\sqrt{m}}$, the following is true for all $\mu \in \Delta_K$:

$$
|\hat{\xi}^{\gamma,t}(\mu) - \xi^t(\mu)| \leq \max_j \sqrt{\frac{v_j(\ln(\sqrt{m}/v_j)+1) + \ln(4N\sqrt{m}/\delta)}{\sqrt{m}}} \tag{52}
$$

Combining with Lemma 3 lets us conclude that, with probability at least $1 - \frac{\delta}{2\sqrt{m}}$, the following is true for all $\mu \in \Delta_K$:

$$
|\hat{\xi}^{\gamma,t}(\mu) - \zeta^{\theta_{1:K}}(\mu, t)| \leq \eta\frac{T_{\mathcal{P}}}{\sqrt{m}} + \max_j \sqrt{\frac{v_j(\ln(\sqrt{m}/v_j)+1) + \ln(4N\sqrt{m}/\delta)}{\sqrt{m}}} \tag{53}
$$

Taking the expectation over $t \sim D_{\mathcal{P}}(t)$, and noting that there are at most $\sqrt{m}$ possible failure events, we can conclude that, with probability at least $1 - \frac{\delta}{2}$, the following is true for all $\phi \in \Phi$:

$$|E_{t \sim D_{\mathcal{P}}(t)}[\hat{\xi}^{\gamma,t}(\phi(t))] - \mathcal{E}_{D_{\mathcal{P}}}(\phi)| \leq \eta \frac{T_{\mathcal{P}}}{\sqrt{m}} + \max_j \sqrt{\frac{v_j(\ln(\sqrt{m}/v_j)+1) + \ln(4N\sqrt{m}/\delta)}{\sqrt{m}}} \quad (54)$$

By combining with Lemma 7, we can conclude that with probability at least $1 - \delta$, the following is true for all $\phi \in \Phi$:

$$
\begin{aligned}
\mathcal{E}_{D_{\mathcal{F}}}(\phi) &\leq E_{t \sim D_{\mathcal{P}}(t)}[\hat{\xi}^{\gamma,t}(\phi(t))] + \max_j \sqrt{\frac{v_j(\ln(\sqrt{m}/v_j)+1) + \ln(4N\sqrt{m}/\delta)}{\sqrt{m}}} \\
&+ \eta \frac{T_{\mathcal{P}}}{\sqrt{m}} + 2(\mathcal{E}_{D_{\mathcal{P}} \bigcup \mathcal{F}}(\phi^*_{D_{\mathcal{P}} \bigcup \mathcal{F}}) - \lambda^*_{D_{\mathcal{P}}}) \\
&+ \hat{d}_{\Phi\Gamma\Phi} + 2(2 + \frac{T_{\mathcal{F}}}{\tau})(\eta \frac{T_{\mathcal{P}}}{\sqrt{m}} + \sqrt{\frac{4v_e(\ln(\sqrt{m}/v_e)+1) + \ln(4\sqrt{m}/\delta)}{\sqrt{m}}}) + \frac{2}{3}\rho T_{\mathcal{F}}^2
\end{aligned}
\quad (55)
$$

We can upper-bound this by adding in an extra $\sqrt{\frac{4v_e(\ln(\sqrt{m}/v_e)+1) + \ln(4\sqrt{m}/\delta)}{\sqrt{m}}}$ term and refactoring to get the bound from the theorem statement. $\qquad\square$

We can derive a somewhat tighter bound by refining this proof, which we will do next.

### D.4    STATEMENT AND PROOF OF THEOREM 2

**Theorem 2.** *Let $\Theta_1, ..., \Theta_{N^{CH}}$ be a finite cover of the convex hull of snapshots $CH(\theta_{1:K}) \subset \mathbb{R}^d$, where $N^{CH} := \lceil (diam(\theta_{1:K})/\gamma)^d \rceil$. Let $v_j^{CH}$ be the VC-dimension of $\Theta_j$, and let $v_e$ be the VC-dimension of $CH(\theta_{1:K})$. Then, with probability at least $1 - \delta$, the following bound holds for all $\phi \in \Phi$:*

$$
\begin{aligned}
\mathcal{E}_{D_{\mathcal{F}}}(\phi) &\leq E_{t \sim D_{\mathcal{P}}(t)}[\hat{\xi}^{\gamma,t}(\phi(t))] + \max_j \sqrt{\frac{v_j^{CH}(\ln(\sqrt{m}/v_j^{CH})+1) + \ln(4N^{CH}\sqrt{m}/\delta)}{\sqrt{m}}} \\
&+ 2(\mathcal{E}_{D_{\mathcal{P}} \bigcup \mathcal{F}}(\phi^*_{D_{\mathcal{P}} \bigcup \mathcal{F}}) - \lambda^*_{D_{\mathcal{P}}}) \\
&+ \hat{d}_{\Phi\Gamma\Phi} + 2(\frac{5}{2} + \frac{T_{\mathcal{F}}}{\tau})(\eta \frac{T_{\mathcal{P}}}{\sqrt{m}} + \sqrt{\frac{4v_e(\ln(\sqrt{m}/v_e)+1) + \ln(4\sqrt{m}/\delta)}{\sqrt{m}}}) + \frac{2}{3}\rho T_{\mathcal{F}}^2
\end{aligned}
\quad (56)
$$

*Proof.* The proof proceeds exactly the same way as the proof of Theorem 1, except that in the step 51 we take the supremum over $\tilde{\theta} \in CH(\theta_{1:K})$ instead of $\tilde{\theta} \in \Theta$. $\qquad\square$

We next prove an alternate bound that may be tighter when $\Phi$ is highly constrained.

### D.5    STATEMENT AND PROOF OF THEOREM 3

**Theorem 3.** *Let $\mathcal{H}_s := \{\theta_{1:K} * \phi : \phi \in \Phi\}$ be a hypothesis class of functions that take $(x, t)$ as input. Let $v_s$ be the VC-dimension of $\mathcal{H}_s$, and let $v_e$ be the VC-dimension of $CH(\theta_{1:K})$. Then, with probability at least $1 - \delta$, the following bound holds for all $\phi \in \Phi$:*

$$
\begin{aligned}
\mathcal{E}_{D_{\mathcal{F}}}(\phi) &\leq \hat{\mathcal{E}}_{D_{\mathcal{P}}}(\phi) + \sqrt{\frac{4v_s(\ln(m/v_s)+1) + \ln(4/\delta)}{m}} \\
&+ 2(\mathcal{E}_{D_{\mathcal{P}} \bigcup \mathcal{F}}(\phi^*_{D_{\mathcal{P}} \bigcup \mathcal{F}}) - \lambda^*_{D_{\mathcal{P}}}) \\
&+ \hat{d}_{\Phi\Gamma\Phi} + 2(\frac{5}{2} + \frac{T_{\mathcal{F}}}{\tau})(\eta \frac{T_{\mathcal{P}}}{\sqrt{m}} + \sqrt{\frac{4v_e(\ln(\sqrt{m}/v_e)+1) + \ln(4\sqrt{m}/\delta)}{\sqrt{m}}}) + \frac{2}{3}\rho T_{\mathcal{F}}^2
\end{aligned}
\quad (57)
$$

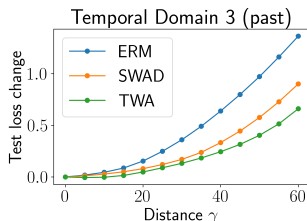 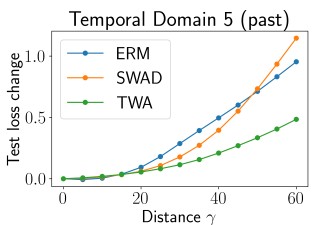 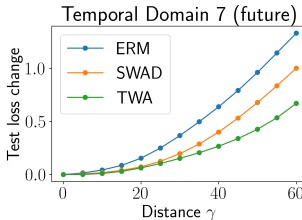

Figure 4: Test loss change $E_{\theta':||\theta'-\theta||\leq\gamma}[\mathcal{E}_{D_i}(\theta') - \mathcal{E}_{D_i}(\theta)]$ as a function of radius $\gamma$. The use of a time-sensitive selector $\phi$ allows TWA to find flatter minima for each temporal domain. Theorem 1 suggests that these lower robust risks should translate to better future generalization.

*Proof.* We start by putting a standard VC-bound on $\mathcal{E}_{D_{\mathcal{P}}}(\phi)$. With probability at least $1 - \frac{\delta}{2}$, the following is true for all $\phi \in \Phi$:

$$|\hat{\mathcal{E}}_{D_{\mathcal{P}}}(\phi) - \mathcal{E}_{D_{\mathcal{P}}}(\phi)| \leq \sqrt{\frac{4v_s(\ln(m/v_s)+1) + \ln(4/\delta)}{m}} \tag{58}$$

Combining with Lemma 7 gets us the conclusion of the theorem. □

### D.6 EMPIRICAL FLATNESS ANALYSIS

To verify whether TWA can find flat minima, where empirical robust risk $E_{t\sim D_{\mathcal{P}}(t)}[\hat{\xi}^{\gamma,t}(\phi(t))]$ is low, we look at the test loss landscape in past and future temporal domains of CLEAR-10. Following Cha et al. (2021), we measure flatness of model $\theta$ on temporal domain $D_i$ by computing expected loss change $E_{\theta':\|\theta'-\theta\|\leq\gamma}[\mathcal{E}_{D_i}(\theta') - \mathcal{E}_{D_i}(\theta)]$ for radius $\gamma$, where $\mathcal{E}_{D_i}(\theta)$ is test loss of $\theta$ on temporal domain $D_i$. We use Monte-Carlo with 50 trials. When evaluating TWA, we use $\theta := \theta_{1:K} * \phi(\bar{t}_i)$ where $\bar{t}_i$ is midpoint time of temporal domain $D_i$. We also evaluate baselines SWAD and ERM, in which $\theta$ is stationary. Fig. 3 shows expected loss change as a function of $\gamma$, and shows that TWA finds flatter minima than the baselines. This means that TWA has lower robust risks, which, per Theorem 1, allows better generalization to future data.

## E ILLUSTRATIVE EXAMPLE OF TWA BENEFITS

In order to better illustrate and understand the benefits of time-sensitive weight averaging, we construct a toy experiment in which the data distribution shifts continuously over time. In this dataset, $x$ is a 2D vector and $y$ is either $+1$ or $-1$.

The distribution is a mixture of Gaussians, where $D(y|h) := \text{Unif}(\{-1, +1\})$ and $D(x|y,h) := y\mu_h + \epsilon$, where $\mu_h := [\cos(2\pi h/H), \sin(2\pi h/H)]$ and $\epsilon \sim \mathcal{N}(0, \sigma^2 I_2)$ is IID spherical Gaussian noise. $D(h|t)$ is periodic and is defined as $\text{softmax}(\frac{1}{\omega}M[\cos(2\pi\text{poly}(t/H)); \sin(2\pi\text{poly}(t/H))])$, where $\text{poly}(\cdot)$ is a polynomial and $M_{h,\cdot} := \mu_h$. For our experiments, we set $H := 12$, $\sigma := 0.5$, $\omega := 0.25$, and poly is a 1-degree polynomial (i.e. linear function) with slope 1.333 and intercept 0.42.

In this experiment, time is always discretized to integer values $0, 1, 2, ..., 11$, where 0 through 8 is the past, and 9 through 11 is the future. 60 datapoints are sampled per time.

Our snapshots $\theta_{1:K}$ are linear models without any bias, and we use $K := 9$ of them. We obtain the snapshots using the following procedure. First, initialize base model by sampling from Gaussian with bandwidth 0.05. Then, optimize for 100k steps, using a batch size of 1 for each step (and sampling with replacement), and minimizing the linear loss $-y\theta x$. Use weight-clipping to constrain the model magnitude to always be $\leq 0.15$. Use a constant learning rate of 0.1 to ensure adequate diversity of snapshots. Sample snapshots uniformly from among the last 10k iterations.

The selector has a similar form to the data-generating model. Its has the following form:

$$\text{softmax}(A[\cos(2\pi\text{poly}(t/H)); \sin(2\pi\text{poly}(t/H))] + b), \tag{59}$$

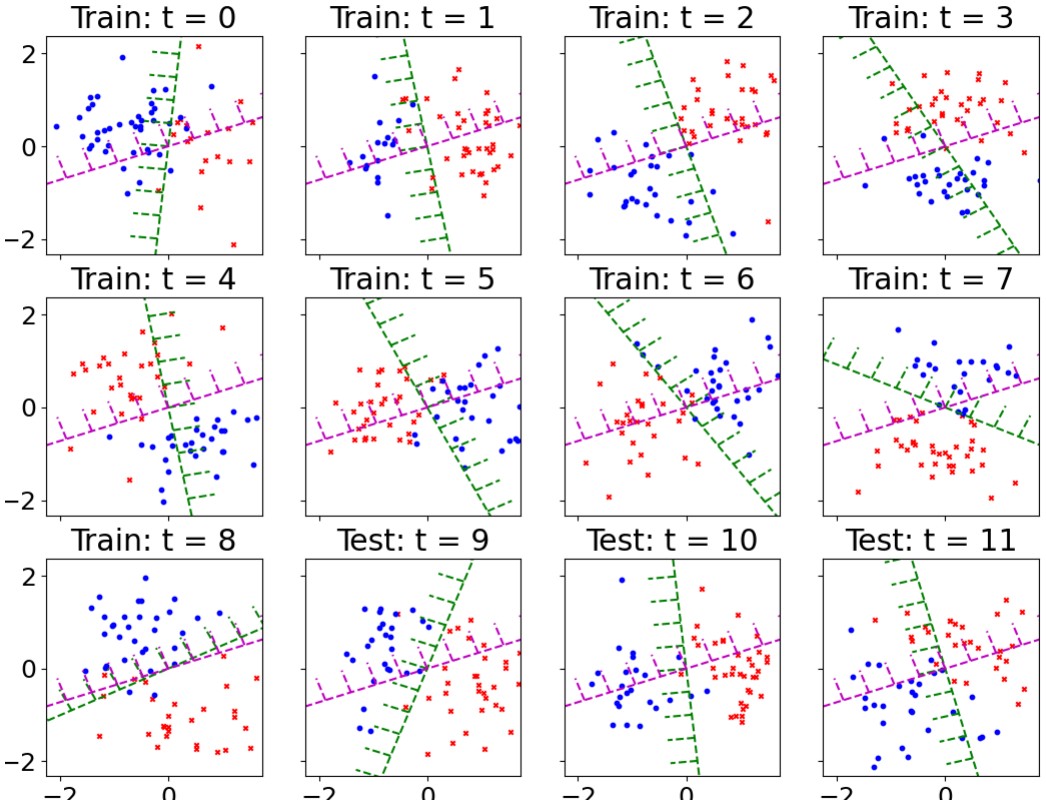

Figure 5: Data and results for one run of the toy experiment. Blue points are positive, red points are negative. Green line is decision boundary for TWA, magenta line is for stationary weight-averaging. The "comb" indicates which side of the decision boundary is predicted positive.

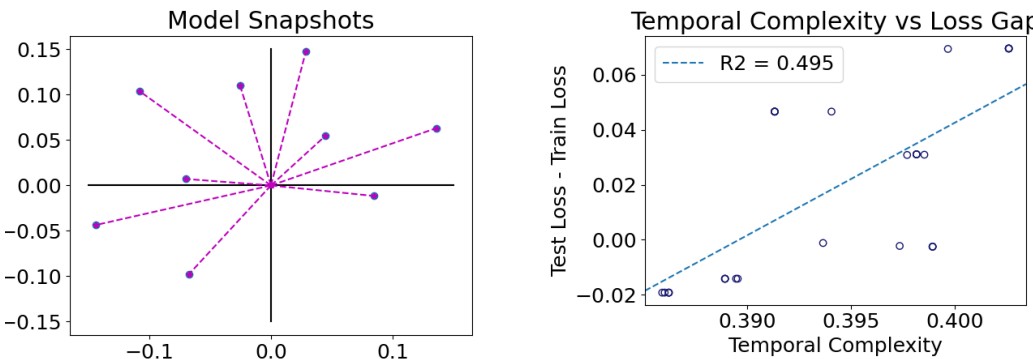

Figure 6: **(left)** Model snapshots from one run of the toy experiment, shown as direction vectors (i.e. decision boundaries would be perpendicular). **(right)** Temporal complexity vs difference between test loss and train loss, with trendline. Higher temporal complexity is positively correlated with a wider gap between test and train loss, in line with our theoretical result.

where $A$, $b$, and polynomial coefficients are learnable. The number of degrees in the polynomial is a hyperparameter. As with the model snapshots, the selector is trained to minimize linear loss $-y(\theta_{1:K} \cdot \phi(t)) \cdot x$, with the constraint that all non-intercept polynomial coefficients are in $[-3, +3]$ interval (the intercept is unconstrained), and all entries in $A$ and $b$ are in $[-1, +1]$ interval. All parameters are initialized uniformly within their constraint bounds, and L-BFGS-B is used to mini-

mize loss across the whole training set at once. We do 100 trials of this minimization and return the selector that got the lowest training loss. For each trial, there is a 0.2 probability that we will blend our initial model with the best one so far, in order to harness both exploration and exploitation.

Fig. 1 shows the data and results for one run of the toy experiment. Positive points are shown in blue, and negatives in red. The decision boundary given by TWA is shown in green, with the "comb" indicating which side of the boundary is predicted positive. As a baseline, we also plot the decision boundary of a stationary uniform average of the model snapshots, in magenta. We see that TWA is able to adapt to the temporal domain shift, while the performance of the stationary average is unreliable due to the shift. The left side of Fig. 2 shows the model snapshots as direction vectors.

We also take a look at temporal complexity $\hat{d}_{\Phi\Gamma\Phi}$ to see how well it correlates with the gap between train and test risks. For ease of optimization, we measure temporal complexity in terms of gaps in linear loss, instead of 0-1 loss, and likewise measure its correlation with the difference between linear test loss and linear train loss. Measuring temporal complexity is similar to training a selector, since it involves maximizing a quantity that is a function of a pair of selectors.

We generate the toy dataset with 5 different random seeds and evaluate the train loss, test loss, and temporal complexity (in terms of linear losses) for selectors with polynomial degrees 1-6, giving us a total of 30 experiments. The right side of Fig. 2 plots temporal complexity on the x-axis and test-loss-minus-train-loss on the y-axis. We see that there is a positive correlation between the two quantities, which is what we would expect from our theoretical result. In particular, the two quantities have a Pearson correlation coefficient of $0.703$, and hence an $R^2$ value of $0.495$, meaning that our temporal complexity explains almost 50% of the variance in the train-test loss gap for these toy experiments.

## F  TWA TRAINING AND INFERENCE ALGORITHMS

In this section, we present pseudocodes for TWA training and inference. We assume that there is a dataset consisting of source temporal domains $D_1, ..., D_s$ and target temporal domains $D_{s+1}, ..., D_E$. Alg. 1 describes how to create model snapshots $\theta_{1:K}$ using source domains $D_1, ..., D_s$, given total iterations $I_\theta$, late-sampling iterations $I_{LS}$, sampling strategy $\mathcal{S}$, and model snapshot number $K$. Alg. 2 describes how to optimize selector $\phi$ on source domains $D_1, ..., D_s$ using snapshots $\theta_{1:K}$ from the previous algorithm, given iterations $I_\phi$. Finally, Alg. 3 describes how to use snapshots $\theta_{1:K}$ and trained selector $\phi$ to obtain a prediction for a query sample $(x_{te}, t_{te})$, which could come from any of the target domains $D_{s+1}, ..., D_E$. Note that $\phi(t)$ denotes the inference of selector model $\phi$ on a timestamp $t$, while $\mathcal{F}(x, \theta)$ denotes the inference of a base model $\theta$ on an input $x$.

---

**Algorithm 1** TWA Training Algorithm - Creating Snapshots

1: INPUT: Source domains $D_1, ..., D_s$.
2: INPUT: Total iterations $I_\theta$, late-sampling iterations $I_{LS}$, sampling strategy $\mathcal{S}$, model snapshot number $K$.
3: **for** $i = 1, 2, ..., I_\theta - I_{LS}$ **do**
4:     Get training sample $(x, \cdot, y)$ from $D_1, ..., D_s$.
5:     Optimize model $\theta$ with $\mathcal{L}(\mathcal{F}(x, \theta), y)$ while also optimizing the BN layers.
6: **end for**
7: **for** $i = I_\theta - I_{LS} + 1, ..., I_\theta$ **do**
8:     Sample model snapshots $\theta_{1:K} \sim \mathcal{S}_{ls}\left(\arg\min_\theta \sum_{j \in [1,s]} \sum_{(X, \cdot, Y) \sim D_j} \mathcal{L}(\mathcal{F}(X, \theta), Y)\right)$
9: **end for**
10: **return** $\theta_{1:K}$

---

---

**Algorithm 2** TWA Training Algorithm - Optimizing Selector

---

1: INPUT: Source domains $D_1, ..., D_s$.
2: INPUT: Model snapshots $\theta_{1:K}$ from Alg. 1, and iterations $I_\phi$.
3: **for** $i = 1, 2, ..., I_\phi$ **do**
4:     Get training sample $(x, t, y)$ from $D_1, ..., D_s$.
5:     Generate the averaging coefficients $\phi(t)$.
6:     Optimize selector $\phi$ with loss $\sum_{i \in [1,s]} \sum_{(x,t,y) \sim D_i} \mathcal{L}\left(\sum_{k=1}^{K} \phi(t)_k \cdot \mathcal{F}(x, \theta_k), \, y\right)$.
7: **end for**
8: **return** $\phi$

---

**Algorithm 3** TWA Inference Algorithm

---

1: INPUT: Query sample $(x_{te}, t_{te})$ from any of target domains $D_{s+1}, ..., D_E$.
2: INPUT: Model snapshots $\theta_{1:K}$ from Alg. 1, and trained selector $\phi$ from Alg. 2.
3: Generate the averaging coefficients $\phi(t_{te})$.
4: Get the inference model weight by averaging the snapshots $\theta_{avg} = \sum_{k=1}^{K} \phi(t_{te})_k \cdot \theta_k$.
5: Inference with $\hat{y} = \mathcal{F}(x_{te}, \theta_{avg})$
6: **return** $\hat{y}$

---

