# OpenReview forum: "Time-sensitive Weight Averaging for Practical Temporal Domain Generalization"
_ICLR.cc/2024/Conference — ICLR 2024 Conference Withdrawn Submission_

### Official Review · Reviewer_c3ap · 2023-10-28

**Soundness:** 3 good
**Presentation:** 3 good
**Contribution:** 3 good
**Rating:** 6
**Confidence:** 4

**Summary:**

The paper studies a specific case of domain adaptation over a sequence of tasks. Authors assume that distribution between consecutive domain change smoothly. Under this assumption, they propose Time-sensitive Weight Averaging, a learning approach based on weight averaging. The authors also present an experimental study of the proposed algorithms.

**Strengths:**

The paper is well written and easy to follow.

The use of weight averages is one of the most common proposals to adapt to changes over time. The paper makes a good review of state-of-the-art methods that allow to understand the contributions of the proposed method.

The idea of using the time stamp is new and very interesting. In addition, the proposed methods provide theoretical guarantees.

The authors perform extensive numerical experiments

**Weaknesses:**

It will be great to know how much this assumption is agreed and violated on individual tasks on experiments.

It would be nice if the theoretical guarantees appeared in the paper instead of in the appendices.

I think one improvement the authors can consider is to add some discussions about the computational complexity and running times.

having explanations about methods used for comparison could  significantly increase the readability.

The authors' assumption about the distribution is the usual assumption made in supervised classification under concept drift (References below). I think the authors should mention that this is a common assumption (gradual drift assumption) in supervised classification under concept drift as well as describe how the methods that make this assumption adapt to changes in the distribution.

Žliobaitė, I. (2010). Learning under concept drift: an overview. arXiv preprint arXiv:1010.4784.

Elwell, R., & Polikar, R. (2011). Incremental learning of concept drift in nonstationary environments. IEEE Transactions on Neural Networks, 22(10), 1517-1531.

**Questions:**

How does the method perform if you use a dataset with tasks in which the distribution does not change? For example MNIST dataset.
Or what happens if the distribution changes too fast?
Could the method work well but the theoretical guarantees would not be true?

Can the method be used on text datasets?The authors use the image datasets from the wild time library. Could the method be extended to the other three datasets?

Can the method be used for any loss function?

---

> ### Author Response · Authors · 2023-11-18
> **Response to Reviewer c3ap**
>
> > It will be great to know how much this assumption is agreed and violated on individual tasks on experiments.
>
> For synthetic datasets like Rotated MNIST, the assumption of smooth distribution shifts holds because distribution shifts on such datasets are controllable. For real-world datasets, the performance variation of the ERM baseline across different domains can serve as an intuitive indicator. If the changes in ERM performance are smooth, it suggests that distribution shifts are also smooth.
>
> > It would be nice if the theoretical guarantees appeared in the paper instead of in the appendices.
>
> Due to page limitations, we had to move the theoretical section to the appendix. We will consider incorporating some key parts into the main text in the revised version.
>
> > I think one improvement the authors can consider is to add some discussions about the computational complexity and running times.
>
> Yes, we discussed efficiency in Table 11 of the paper. We further add more efficiency and comparisons in the following table.
>
> |      Dataset     |   CLEAR-10  |              |  CLEAR-100  |              |     FMoW    |              |
> |:----------------:|:-----------:|:------------:|:-----------:|:------------:|:-----------:|:------------:|
> |                  | Accuracy(%) | Cost (hours) | Accuracy(%) | Cost (hours) | Accuracy(%) | Cost (hours) |
> |       CIDA       |     87.3    |      6.8     |     69.2    |     14.7     |      ×      |       ×      |
> |     AdaGraph     |     75.5    |      3.4     |     50.2    |     10.3     |      ×      |       ×      |
> | GI |     82.2    |      3.5     |     54.3    |     14.6     |      ×      |       ×      |
> |       LSSAE      |      86     |     17.5     |     62.1    |     58.3     |     58.7    |     43.2     |
> |        DDA       |     86.3    |     15.6     |     41.2    |     42.3     |     49.2    |     33.5     |
> |        ERM       |     86.3    |      1.7     |     68.3    |      8.2     |     63.5    |      4.2     |
> |        TWA       |     88.5    |      1.9     |     72.1    |      8.4     |     69.5    |      5.3     |
>
>
> > Having explanations about methods used for comparison could significantly increase the readability.
>
> Thank you for the suggestion; we will attempt to incorporate more detailed descriptions of the baselines in the revised version.
>
> > The authors' assumption about the distribution is the usual assumption made in supervised classification under concept drift (References below). I think the authors should mention that this is a common assumption (gradual drift assumption) in supervised classification under concept drift as well as describe how the methods that make this assumption adapt to changes in the distribution.
>
> Thank you for the suggestion. We will consider including more discussions regarding concept drift.
>
> > How does the method perform if you use a dataset with tasks in which the distribution does not change? For example MNIST dataset. Or what happens if the distribution changes too fast? Could the method work well but the theoretical guarantees would not be true?
>
> We will add the corresponding ablation study in the revised version. In theory, even if the distribution does not change, TWA can still improve performance. This is because weight averaging itself is a commonly used technique to enhance the generalization ability.
>
> > Can the method be used on text datasets? The authors use the image datasets from the wild time library. Could the method be extended to the other three datasets?
>
> Yes, it can be used on text datasets. In the Section C.2 of appendix, we simply evaluated TWA on 2 text datasets, arXiv and HuffPost. The results show that TWA can also boost generalization ability on text datasets.
>
> > Can the method be used for any loss function?
>
> In theory, TWA can be combined with various loss functions. However, it's unclear how different loss functions might impact the performance of TWA.

---

### Official Review · Reviewer_Jcbo · 2023-10-29

**Soundness:** 3 good
**Presentation:** 3 good
**Contribution:** 3 good
**Rating:** 6
**Confidence:** 4

**Summary:**

This article addresses the issue of enhancing the practicality of Temporal Domain Generalization (TDG). This paper comprehensively considers various aspects of TDG tasks, including methods, datasets, evaluation settings, comparisons with previous methods and more.
It makes the following contributions:
- It analyzes the limitations of previous TDG approaches and summarizes three design principles for TDG methods: 1) Time-sensitive model, 2) Generic method, and 3) Realistic evaluation.
- In the context of TDG tasks, the article improves the Weight Averaging method by automatically learning the optimal weight averaging strategy for different time intervals.
- The authors enhance the TDG benchmark by incorporating more realistic datasets, namely CLEAR-10, CLEAR-100, Yearbook, and FMoW-Time, into the TDG setting.
- Comprehensive evaluation results are provided, including comparison with other methods using both new and previous datasets, and results on both vision and language tasks.

**Strengths:**

- This paper provides valuable analysis, summary, and design principles for improving the practicality of Temporal Domain Generalization. The proposed three design principles are a meaningful step towards enhancing the practicality of TDG.
- This paper proposed a novel TDG method, TWA, which is the first to apply weight averaging toTDG.  And TWA further makes the weight averaging learnable to adapt to the smooth distribution shift setting of TDG.
- This paper is the first one to evaluate TDG methods on larger and more complex datasets, which contributes to the overall improvement of TDG's practicality in complex application scenarios.  Evaluation across new and common TDG benchmarks, along with efficiency assessments, allows us to obtain a comprehensive assessment of the proposed method.
- Strong performance. Comprehensive experiments are conducted, outperforming other methods across various benchmarks and tasks with good efficiency. The proposed TWA is a "simple yet effective" method that can be easily applied and adapted to different tasks, models, and datasets.

**Weaknesses:**

Like most other TDG works, TWA is also limited by some common problems within the TDG task:
- TWA also relies on the “smooth distribution shift” assumption, which could limit its potential applications.
- Evaluations are limited to relatively “simple” tasks, e.g. most TDG methods are evaluated with classification and regression tasks only. It’s unclear whether these TDG methods could still work when generalized to more complex tasks, such as image segmentation or common object detection.

**Questions:**

Overall, I have a positive impression of this paper. The limitations I listed are common for most existing TDG works, which do not hinder TWA from being a novel TDG method. And I got questions about the implementation details:

- When you are applying TWA with different tasks, datasets or models, are you using a constant “number of snapshots”? If not, is there any strategy to select the best number of snapshots for different application settings? It would be helpful if there could be results of “number of snapshots vs. task complexity”.

---

> ### Author Response · Authors · 2023-11-18
> **Response to Reviewer Jcbo**
>
> > TWA also relies on the “smooth distribution shift” assumption, which could limit its potential applications.
>
> Yes, TWA also follows the “smooth distribution shift” assumption of TDG tasks.
>
> > Evaluations are limited to relatively “simple” tasks, e.g. most TDG methods are evaluated with classification and regression tasks only. It’s unclear whether these TDG methods could still work when generalized to more complex tasks, such as image segmentation or common object detection.
>
> Moving from simple datasets and network structures to realistic datasets with larger networks is the first step in making TDG practical. We will explore the potential applications of TDG in various more complex tasks in future work.
>
> > When you are applying TWA with different tasks, datasets or models, are you using a constant “number of snapshots”? If not, is there any strategy to select the best number of snapshots for different application settings? It would be helpful if there could be results of “number of snapshots vs. task complexity”.
>
> For different tasks, we may use different numbers of snapshots. We will add the specific numbers of snapshots to the experimental details.
>
> Based on our experience, it is better to use more snapshots when dealing with complex tasks. Conversely, selecting too many snapshots on simple tasks may harm performance.
>
> As for the  “number of snapshots vs. task complexity”, we refer to Fig.3 (b) to show how performance changes with the number of snapshots on a relatively complex task. We further vary the number of snapshots on the Elec2 dataset, which is a relatively simple task, and get the following results.
>
> | Number of snapshots |      2     |      3     |      5     |
> |:-------------------:|:----------:|:----------:|:----------:|
> |     Accuracy (%)    | 86.5 ± 1.1 | 84.8 ± 2.0 | 83.2 ± 1.7 |

---

### Official Review · Reviewer_Zoin · 2023-10-29

**Soundness:** 3 good
**Presentation:** 2 fair
**Contribution:** 3 good
**Rating:** 3
**Confidence:** 4

**Summary:**

Temporal Domain Generalization (TDG) aims to tackle temporal distribution shifts in models without future sample access. Traditional TDG methods, often oversimplified, were either time-sensitive or tried to estimate optimal model parameters for each temporal domain. To improve TDG's practical applicability, the authors introduce three main principles: a time-sensitive model, a generic method, and realistic evaluation. Following these principles, they present Time-sensitive Weight Averaging (TWA), which uses weight averaging of specialists for every temporal domain and a selector network trained on timestamps. TWA's effectiveness is confirmed through experiments on various benchmarks, showing up to a 4% improvement in accuracy over traditional methods.

**Strengths:**

1. The authors emphasize the importance of addressing Temporal Domain Generalization (TDG) in a more comprehensive manner. They introduce three key principles for TDG method design, which ensure that the approach is time-sensitive, generic, and evaluated under realistic conditions.
2. The TWA approach is simple and effective. It leverages weight averaging for every temporal domain and is complemented by a selector network that estimates the best coefficients based on timestamp input.
3. TWA has been tested extensively on multiple realistic benchmarks and demonstrated great performance.

**Weaknesses:**

1. The motivation is a little weak in my opinion. Specifically, the authors criticize that existing methods rely too much on time-sensitive mechanisms, which may cause troubles for large generic models. However, given the task is **temporal** DG, it is quite intuitive to rely on time-sensitive information, and authors seem to agree with the strength of using the time-sensitive information. For most of the generic models like ChatGPT, the problem of DG may not be a big deal. Applying weight averaging to these foundation models is also a resource-consuming task.

2. The technical design is a little trivial and odd to me. The motivation says existing methods rely too much on temporal information, we should use weight averaging instead. However, the proposed method still seems to leverage a selector network to model the trajectory of the distribution shift over time. In addition, the model is built upon SWAD (Cha et al., 2021) with some additional contributions to learning the averaging coefficient.

3. Finally, the presentation of the manuscript still has a large room for improvement. For example, all the notations in Figure 1 are not introduced, which makes the visual presentation not self-contained. In addition, some sentences are hard to understand, e.g., "They assumed that understanding say how a mobile phone’s appearance has changed in the past may help predict future changes."

**Questions:**

1. What is the definition of model snapshots? Is it the same thing as the model parameters?
2. Following the previous question, if the model snapshot equals the model parameter, how can it be sampled? Is the model a Bayesian Neural Network?
3. One major contribution that the authors claim is "The method can be easily combined with various architectures and tasks, requiring as few architecture modifications as possible". Therefore, it makes people expect there may be some experiments showing the proposed method is flexible with different architectures. In addition, runtime comparisons with other methods can also be demonstrated. However, Table 11 only contains the runtime comparison against itself.
4. In the experiments, why the model is heavily compared with many DG methods, like ERM, MTL, GROUPDRO, etc? The major comparison between the proposed method and TDG methods is better to be placed in the main content instead of the appendix.
5. The most up-to-date TDG method DRAIN is not compared in many datasets. Could the authors explain what exactly the "highly task-specific design" is in DRAIN (Bai et al., 2022)?

---

> ### Author Response · Authors · 2023-11-18
> **Response to Reviewer Zoin**
>
> > The motivation is a little weak in my opinion. Specifically, the authors criticize that existing methods rely too much on time-sensitive mechanisms, which may cause troubles for large generic models. However, given the task is temporal DG, it is quite intuitive to rely on time-sensitive information, and authors seem to agree with the strength of using the time-sensitive information. For most of the generic models like ChatGPT, the problem of DG may not be a big deal. Applying weight averaging to these foundation models is also a resource-consuming task.
>
> We need to clarify that we are not criticizing using time-sensitive mechanisms. In fact, we also believe that time-sensitive mechanisms are essential for TDG tasks. Our difference in motivation from previous works lies in that, previous works primarily concentrated on improving generalization performance through better utilization of temporal information, and we aim at incorporating more practical considerations into the design and evaluation of TDG. And we believe these 2 motivations are equally important.
>
> The major issue we are addressing is that the time-sensitive mechanisms within many existing methods cannot generalize well to larger and more realistic application scenarios. As demonstrated in Table 1 of our global response, they either result in a severe performance drop or lead to a significant increase in training costs. These results also underscore the importance of considering TDG issues from a practical view.
>
> While large models trained on large datasets may directly achieve strong generalization, there are many other application scenarios that models and dataset are not “large” enough to make DG/TDG less necessary. However, these scenarios could still be larger in scale than those studied in previous TDG approaches. Therefore, incorporating more practical considerations into TDG is a necessary step in bridging TDG with real-world applications.
>
> > The technical design is a little trivial and odd to me. The motivation says existing methods rely too much on temporal information, we should use weight averaging instead. However, the proposed method still seems to leverage a selector network to model the trajectory of the distribution shift over time. In addition, the model is built upon SWAD (Cha et al., 2021) with some additional contributions to learning the averaging coefficient.
>
> Rather than opposing using temporal information, we are actually suggesting to improve the way to use temporal information from a practical perspective, making TDG methods more generic, easier to be applied to various application scenarios. And the weight averaging method exhibited the desired properties, so we chose to combine the weight averaging method with the temporal mechanism.
>
> In the global response, we reiterated our novelty and contribution. When specifically comparing with SWAD, we drew inspiration from SWAD in the aspects of theoretical guarantees and overfit-aware sampling scheduling. And the other parts, including model training (e.g. normalization), model averaging approach (dense or not), and sampling strategy for TDG (late sampling) are totally different from SWAD.
>
> > Finally, the presentation of the manuscript still has a large room for improvement. For example, all the notations in Figure 1 are not introduced, which makes the visual presentation not self-contained. In addition, some sentences are hard to understand, e.g., "They assumed that understanding say how a mobile phone’s appearance has changed in the past may help predict future changes."
>
> Thank you for pointing out these areas for improvement. We will refine these details in the revised version to ensure clearer expressions.

---

> > ### Author Response · Authors · 2023-11-18
> > **Response to Reviewer Zoin**
> >
> > > What is the definition of model snapshots? Is it the same thing as the model parameters?
> >
> > Yes, snapshot refers to model parameters / weights.
> >
> > > Following the previous question, if the model snapshot equals the model parameter, how can it be sampled? Is the model a Bayesian Neural Network?
> >
> > It’s not a Bayesian Neural Network. “Sample” here just means “saving the model weights at a specific moment / iteration.” And this is the third part where we “follow” SWAD.
> >
> > > One major contribution that the authors claim is "The method can be easily combined with various architectures and tasks, requiring as few architecture modifications as possible". Therefore, it makes people expect there may be some experiments showing the proposed method is flexible with different architectures. In addition, runtime comparisons with other methods can also be demonstrated. However, Table 11 only contains the runtime comparison against itself.
> >
> > We agree that demonstrating genericity with different architectures is important.
> > We have covered a portion by showcasing the performance of combining TWA with various network architectures on different benchmarks. In all our experiments, we applied TWA to 3 different MLPs, 3 different CNNs, and 1 transformer architecture. TWA consistently demonstrated improvements in temporal generalization ability. However, other forms of generality, such as requiring less additional implementation and reduced tuning when integrated with new network architectures, may be challenging to quantitatively demonstrate.
> >
> > As for more efficiency evaluation and comparison with other TDG methods, we add them into the following table.
> >
> > |      Dataset     |   CLEAR-10  |              |  CLEAR-100  |              |     FMoW    |              |
> > |:----------------:|:-----------:|:------------:|:-----------:|:------------:|:-----------:|:------------:|
> > |                  | Accuracy(%) | Cost (hours) | Accuracy(%) | Cost (hours) | Accuracy(%) | Cost (hours) |
> > |       CIDA       |     87.3    |      6.8     |     69.2    |     14.7     |      ×      |       ×      |
> > |     AdaGraph     |     75.5    |      3.4     |     50.2    |     10.3     |      ×      |       ×      |
> > | GI |     82.2    |      3.5     |     54.3    |     14.6     |      ×      |       ×      |
> > |       LSSAE      |      86     |     17.5     |     62.1    |     58.3     |     58.7    |     43.2     |
> > |        DDA       |     86.3    |     15.6     |     41.2    |     42.3     |     49.2    |     33.5     |
> > |        ERM       |     86.3    |      1.7     |     68.3    |      8.2     |     63.5    |      4.2     |
> > |        TWA       |     88.5    |      1.9     |     72.1    |      8.4     |     69.5    |      5.3     |
> >
> > > In the experiments, why the model is heavily compared with many DG methods, like ERM, MTL, GROUPDRO, etc? The major comparison between the proposed method and TDG methods is better to be placed in the main content instead of the appendix.
> >
> > We involve DG baselines to enhance the comprehensiveness of the comparisons, following DDA's experimental settings. We will consider moving the comparisons with DRAIN from the appendix to the main text, replacing those DG baselines.
> >
> > > The most up-to-date TDG method DRAIN is not compared in many datasets. Could the authors explain what exactly the "highly task-specific design" is in DRAIN (Bai et al., 2022)?
> >
> > "Task-specific implementation" is more suitable. Methodologically, DRAIN provides a generic framework and can integrate with various model architectures. But the official implementation of DRAIN is a little bit specific to the network architectures used in the paper. When it comes to a new architecture, we need to implement a new construction function. We tried to adapt DRAIN to our new settings, e.g. CLEAR-10 and ResNet-18, but cannot make DRAIN learn properly.
> >
> > As comparisons with DRAIN are not available on our new datasets, we compare TWA with DRAIN under its own experimental settings as an alternative. We expanded the comparison with DRAIN to 3 datasets. From the results, it can be seen that our TWA achieves comparable performance to DRAIN on Rot-MNIST and Elec2, while being notably better on ONP.
> >
> > |   Method  |  Rot-MNIST |     ONP    |    Elec2   |
> > |:---------:|:----------:|:----------:|:----------:|
> > |    ERM    | 81.4 ± 4.0 | 66.2 ± 0.6 | 77.0 ± 3.1 |
> > |    CDOT   | 85.8 ± 1.0 | 65.9 ± 0.0 | 82.2 ± 0.6 |
> > |    CIDA   | 90.7 ± 0.7 | 65.3 ± 0.6 | 85.9 ± 0.2 |
> > |     GI    | 92.3 ± 1.3 | 63.6 ± 0.8 | 83.1 ± 0.7 |
> > |   DRAIN   | 92.5 ± 1.1 | 61.7 ± 1.2 | 87.3 ± 0.8 |
> > |    DDA    | 91.3 ± 1.5 | 66.7 ± 0.3 | 86.1 ± 0.9 |
> > | DDA + TWA | 92.7 ± 0.7 | 67.0 ± 0.3 | 86.5 ± 1.1 |

---

### Official Review · Reviewer_bw8U · 2023-11-01

**Soundness:** 2 fair
**Presentation:** 3 good
**Contribution:** 2 fair
**Rating:** 3
**Confidence:** 4

**Summary:**

This paper extends weight averaging technique for domain generalization in non-stationary environments. Specifically, the authors employ the Time2Vec model to compute time-sensitive weights, facilitating the creation of an optimal model at each time step by leveraging existing model snapshots. Experimental findings across various real-world datasets consistently showcase the superior performance of the proposed approach compared to baseline methods.

**Strengths:**

- This paper tackles an important issue of temporal domain generalization in machine learning under distribution shift.

- The experiment covers more real-world dataset compared to existing works for temporal domain generalization.

- Code is made available for the purpose of ensuring reproducibility.

**Weaknesses:**

- The authors assert that they introduce a comprehensive benchmark, encompassing several realistic datasets and baselines for temporal domain generalization. However, the comparison with existing temporal domain generalization methods is conducted solely on the Portrait dataset. For other real datasets such as CLEAR, FMoW, arXiv, and HuffPost, the authors only provide comparisons with the weight-averaging method (SWAD [1]). It is recommended that the authors expand their analysis to include results from other methods, particularly those tailored for temporal domain generalization (e.g., GI [2], LSSAE [3], DDA [4], DRAIN [5], DPNET [6]), on these datasets to bolster the strength of their contribution. (NOTE: This is a significant concern. I am open to revising my evaluation if the authors can address and resolve this issue.)

- The proposed method does not strictly adhere to the problem formulation outlined in Equation (1). Notably, while the original problem is cast as a bi-level optimization, the proposed algorithm utilizes a two-phase training approach to optimize the two terms sequentially. The authors should clarify the rationale behind their model design.

- The two-stage training approach employed by the proposed method raises concerns about both time (for training the selector network) and space (for storing model snapshots) complexity.

- The explanation of how the theoretical results influence the development of the proposed method is somewhat lacking in clarity. From my point of view, the current theoretical findings appear to be more apt for providing a theoretical assurance of TWA rather than serving as the primary guiding influence on the algorithm's design.

- Could the authors offer more detailed insights to distinguish their work from SWAD? From my point of view, the technical contribution seems somewhat limited. It appears that the primary contribution lies in the utilization of Time2Vec to adapt the weight-averaging method to the temporal-shift scenario.

References:

[1] Cha, Junbum, et al. "Swad: Domain generalization by seeking flat minima." Advances in Neural Information Processing Systems 34 (2021): 22405-22418.

[2] Nasery, Anshul, et al. "Training for the future: A simple gradient interpolation loss to generalize along time." Advances in Neural Information Processing Systems 34 (2021): 19198-19209.Nasery, Anshul, et al. "Training for the future: A simple gradient interpolation loss to generalize along time." Advances in Neural Information Processing Systems 34 (2021): 19198-19209.

[3] Qin, Tiexin, Shiqi Wang, and Haoliang Li. "Generalizing to Evolving Domains with Latent Structure-Aware Sequential Autoencoder." International Conference on Machine Learning. PMLR, 2022.

[4] Zeng, Qiuhao, et al. "Foresee What You Will Learn: Data Augmentation for Domain Generalization in Non-Stationary Environments." arXiv preprint arXiv:2301.07845 (2023).

[5] Bai, Guangji, Chen Ling, and Liang Zhao. "Temporal Domain Generalization with Drift-Aware Dynamic Neural Networks." arXiv preprint arXiv:2205.10664 (2022).

[6] Wang, William Wei, et al. "Evolving Domain Generalization." arXiv preprint arXiv:2206.00047 (2022).

**Questions:**

Questions are given in the Weaknesses.

---

> ### Author Response · Authors · 2023-11-18
> **Response to Reviewer bw8U**
>
> > The authors assert that they introduce a comprehensive benchmark, encompassing several realistic datasets and baselines for temporal domain generalization. However, the comparison with existing temporal domain generalization methods is conducted solely on the Portrait dataset. For other real datasets such as CLEAR, FMoW, arXiv, and HuffPost, the authors only provide comparisons with the weight-averaging method (SWAD [1]). It is recommended that the authors expand their analysis to include results from other methods, particularly those tailored for temporal domain generalization (e.g., GI [2], LSSAE [3], DDA [4], DRAIN [5], DPNET [6]), on these datasets to bolster the strength of their contribution. (NOTE: This is a significant concern. I am open to revising my evaluation if the authors can address and resolve this issue.)
>
> We appreciate your suggestions. In the following, we are glad to provide more results of TDG baselines, the reason why some results are not available and our motivation to split the comparison into 2 parts.
>
> ### &nbsp;&nbsp; Accuracy and efficiency evaluation on CLEAR-10 and CLEAR-100
>
> First, we evaluate various TDG methods on CLEAR-10 and CLEAR-100. The last domain is used as the target domain. These methods include CIDA, AdaGraph, GI, DDA, and LSSAE. The results are presented in the following Table. It can be observed that TWA exhibits a more pronounced advantage when compared to these TDG methods on CLEAR-10 and CLEAR-100.
>
> |      Dataset     |   CLEAR-10  |              |  CLEAR-100  |              |     FMoW    |              |
> |:----------------:|:-----------:|:------------:|:-----------:|:------------:|:-----------:|:------------:|
> |                  | Accuracy(%) | Cost (hours) | Accuracy(%) | Cost (hours) | Accuracy(%) | Cost (hours) |
> |       CIDA       |     87.3    |      6.8     |     69.2    |     14.7     |      ×      |       ×      |
> |     AdaGraph     |     75.5    |      3.4     |     50.2    |     10.3     |      ×      |       ×      |
> | GI |     82.2    |      3.5     |     54.3    |     14.6     |      ×      |       ×      |
> |       LSSAE      |      86     |     17.5     |     62.1    |     58.3     |     58.7    |     43.2     |
> |        DDA       |     86.3    |     15.6     |     41.2    |     42.3     |     49.2    |     33.5     |
> |        ERM       |     86.3    |      1.7     |     68.3    |      8.2     |     63.5    |      4.2     |
> |        TWA       |     88.5    |      1.9     |     72.1    |      8.4     |     69.5    |      5.3     |
>
> ### &nbsp;&nbsp; Missing Comparison with GI and DRAIN
>
> **GI** is hard to be adapted to large datasets due to the gradient interpolation loss. Training a network with TReLU is feasible in terms of efficiency and accuracy. However, simply using GI loss leads to a significant performance drop. While finding adaptation with extensive experiments might alleviate this issue, the high cost of GI loss prevents us from this. On CLEAR-10, a single finetuning epoch with GI loss requires 23 GPU hours, and on CLEAR-100, each epoch even demands 153 GPU hours.
>
> **DRAIN** is missing as we cannot facilitate its proper training on new benchmarks. Methodologically, DRAIN is generic and can easily integrate with various model architectures. However, its implementation and empirical evaluation are primarily designed for small datasets, such as MNIST, and small models like MLP or 2-layer CNN. When we attempt to apply DRAIN to larger datasets, such as CLEAR, and larger CNN networks, such as ResNet, we find it hard to make the network learn properly, resulting in random guessing networks.
>
> ### &nbsp;&nbsp; 2 parts of comparisons
>
> When many TDG methods struggle to generalize to the new settings, comparing with them only using new benchmarks and TDG methods may raise concerns about unfair comparisons. Perhaps better method adaptations could enhance the accuracy. But the significant cost increases are basically unavoidable, which also makes it hard to explore better adaptations. Therefore, we additionally selected some datasets commonly used by these TDG methods to enhance the fairness and comprehensiveness of the evaluation.
>
> ### &nbsp;&nbsp; Other relevant clarifications.
>
> **CIDA and AdaGraph** are used as TDG baselines, as they are also selected as baselines by most other TDG methods. And TDG can be regarded as a special case of their original Continuous Domain Adaptation task.
>
> **ArXiv and HuffPost** are relatively less crucial, as they are simply used to test whether TWA can generalize to NLP tasks, instead of serving as the major benchmarks. And they also require using pretrained weights, making many TDG methods cannot be evaluated.
>
> **DPNET** seems to be not published yet.

---

> ### Author Response · Authors · 2023-11-18
> **Response to Reviewer bw8U**
>
> > The proposed method does not strictly adhere to the problem formulation outlined in Equation (1). Notably, while the original problem is cast as a bi-level optimization, the proposed algorithm utilizes a two-phase training approach to optimize the two terms sequentially. The authors should clarify the rationale behind their model design.
>
> We provide Eq.1 as a way to introduce the TDG formulationally generally. However, Eq. 3 represents the specific form of Eq. 1 that we used in our paper.
>
> > The two-stage training approach employed by the proposed method raises concerns about both time (for training the selector network) and space (for storing model snapshots) complexity.
>
> According to Table 11, we can observe that TWA does not significantly increase training time, only resulting in an increase of 2-20% of the total cost. As for the storage of snapshots, on one hand, the memory consumption of model weights is significantly smaller than that of feature maps. On the other hand, in practical applications, we can categorize a period of time into a domain, where all data within that period share the same domain index. This allows us to directly combine weights to obtain a single averaged model to save memory. For instance, store all snapshots and selectors on servers and use the single averaged model on memory-constrained devices.
>
> > The explanation of how the theoretical results influence the development of the proposed method is somewhat lacking in clarity. From my point of view, the current theoretical findings appear to be more apt for providing a theoretical assurance of TWA rather than serving as the primary guiding influence on the algorithm's design.
>
> Yes, in the Section 3.3, we are stating ”we have derived some **theoretical guarantees** on the ability of TWA to generalize from past to future data”
>
> > Could the authors offer more detailed insights to distinguish their work from SWAD? From my point of view, the technical contribution seems somewhat limited. It appears that the primary contribution lies in the utilization of Time2Vec to adapt the weight-averaging method to the temporal-shift scenario.
>
> In the global response, we reiterated our novelty and contribution. When specifically comparing with SWAD, we drew inspiration from SWAD in the aspects of theoretical guarantees and overfit-aware sampling scheduling. And the other parts, including model training (e.g. normalization), model averaging approach (dense or not), and sampling strategy for TDG (late sampling) are totally different from SWAD. We also want to argue that we are specifically targeting the TDG task, and excluding the TDG-specific algorithm designs and theoretical analysis from technical contributions is unreasonable and not objective.

---

### Author Response · Authors · 2023-11-18
**Global Rebuttal**

We greatly appreciate the constructive comments provided by the reviewers. In this response, we address some common concerns.

## 1. Contributions

The fundamental contribution of this paper lies in providing a systematic summary and analysis of TDG tasks from a practical perspective. Through synthesizing real-world application requirements and limitations of existing methods, we introduce the three guiding principles for TDG design. This is rarely touched by prior TDG works, and constitutes a significant novel contribution of this paper.

Introducing TWA is, of course, another important contribution, which has the following key benefits over prior work:

* As far as we know, we are the first to apply the Model Averaging method to TDG tasks. This was not a straightforward endeavor, as TDG tasks exhibit many differences from DG tasks For example, in the context of TDG, it is essential to focus more on the smooth distribution shift nature, as well as exploring strategies to leverage the domain index. We conducted essential adaptations during this process, including normalization and sampling strategies.
* Making the Model Averaging method time-sensitive itself is inherently novel, as no prior work has done this. Moreover, within TWA, we took further steps to tailor the design to the characteristics of TDG tasks, incorporating features like Time2Vec and optimization strategies. While it is true that we drew inspiration from SWAD in various aspects, they do not constitute a direct adaptation of their approach.  Instead, the improvements we made are far more significant for TDG tasks.
* Simplicity is a key aspect of our novelty. Guided by generic design principles, the simplicity of TWA is an intentional attribute, which allows for natural integration with various network architectures, tasks, and even other DG/TDG methods. For example, when integrated with ResNet, AdaGraph requires changing all BN layers to AdaBN, while TWA requires no change to the architecture.

Last but not least, our novelty also lies in the new evaluation settings. This work is the first to incorporate large datasets with natural distribution shifts as realistic TDG benchmarks. Compared to prior work [33, 35, 53, 58], our TDG evaluation procedures in this paper better reflect real-world TDG applications.



## 2. Evaluation of TDG methods on CLEAR-10, CLEAR-100 and FMoW

|      Dataset     |   CLEAR-10  |              |  CLEAR-100  |              |     FMoW    |              |
|:----------------:|:-----------:|:------------:|:-----------:|:------------:|:-----------:|:------------:|
|                  | Accuracy(%) | Cost (hours) | Accuracy(%) | Cost (hours) | Accuracy(%) | Cost (hours) |
|       CIDA       |     87.3    |      6.8     |     69.2    |     14.7     |      ×      |       ×      |
|     AdaGraph     |     75.5    |      3.4     |     50.2    |     10.3     |      ×      |       ×      |
| GI |     82.2    |      3.5     |     54.3    |     14.6     |      ×      |       ×      |
|       LSSAE      |      86     |     17.5     |     62.1    |     58.3     |     58.7    |     43.2     |
|        DDA       |     86.3    |     15.6     |     41.2    |     42.3     |     49.2    |     33.5     |
|        ERM       |     86.7    |      1.6     |     68.3    |      8.0     |     63.5    |      4.2     |
|        TWA       |     88.5    |      1.9     |     72.1    |      8.4     |     69.5    |      5.3     |


## 3. More comparisons with DRAIN

|   Method  |  Rot-MNIST |     ONP    |    Elec2   |
|:---------:|:----------:|:----------:|:----------:|
|    ERM    | 81.4 ± 4.0 | 66.2 ± 0.6 | 77.0 ± 3.1 |
|    CDOT   | 85.8 ± 1.0 | 65.9 ± 0.0 | 82.2 ± 0.6 |
|    CIDA   | 90.7 ± 0.7 | 65.3 ± 0.6 | 85.9 ± 0.2 |
|     GI    | 92.3 ± 1.3 | 63.6 ± 0.8 | 83.1 ± 0.7 |
|   DRAIN   | 92.5 ± 1.1 | 61.7 ± 1.2 | 87.3 ± 0.8 |
|    DDA    | 91.3 ± 1.5 | 66.7 ± 0.3 | 86.1 ± 0.9 |
| DDA + TWA | 92.7 ± 0.7 | 67.0 ± 0.3 | 86.5 ± 1.1 |